# Demystifying the Mechanisms Behind Emergent Exploration in Goal-conditioned RL

**Mahsa Bastankhah**[*]
Department of Electrical and Computer Engineering
Princeton University
mb6458@princeton.edu

**Grace Liu**[*]
Machine Learning Department
Carnegie Mellon University
gliu2@andrew.cmu.edu

**Dilip Arumugam**[1]**, Thomas L. Griffiths**[1,2]**, Benjamin Eysenbach**[1]
[1]Department of Computer Science
[2]Department of Psychology
Princeton University

## Abstract

In this work, we take a first step toward elucidating the mechanisms behind emergent exploration in unsupervised reinforcement learning. We study Single-Goal Contrastive Reinforcement Learning (SGCRL) (Liu et al., 2025), a self-supervised algorithm capable of solving challenging long-horizon goal-reaching tasks without external rewards or curricula. We combine theoretical analysis of the algorithm's objective function with controlled experiments to understand what drives its exploration. We show that SGCRL maximizes implicit rewards shaped by its learned representations. These representations automatically modify the reward landscape to promote exploration before reaching the goal and exploitation thereafter. Our experiments also demonstrate that these exploration dynamics arise from learning low-rank representations of the state space rather than from neural network function approximation. Our improved understanding enables us to adapt SGCRL to perform safety-aware exploration.

## 1 Introduction

Recent breakthroughs in deep reinforcement learning (RL) have revealed emergent behaviors: agents that develop skills without explicit rewards (Liu et al., 2025), learn to plan without world models (Bush et al., 2025; Simmons-Edler et al., 2025), and exhibit sophisticated exploration in open-ended environments (Team et al., 2021). For example, Single-Goal Contrastive Reinforcement Learning (SGCRL), learns several manipulation skills before receiving any rewards and without any explicit skill-learning objectives (Liu et al., 2025). Subsequent work has demonstrated this phenomenon in robotic manipulation (Liu et al., 2025), locomotion (Bortkiewicz et al., 2025), and multi-agent tasks (Nimonkar et al., 2025) (see Fig. 1), but the underlying mechanism that drives this emergent exploration remains unknown.

Conventional wisdom suggests that emergent properties arise from using large models (Bubeck et al., 2023), but even small neural networks learn feature hierarchies (Krizhevsky et al., 2009), and learning word representations that support analogical reasoning is a consequence of the choice of loss function, rather than architecture (Hashimoto et al., 2016; Arora et al., 2016). In the context of reinforcement learning, the extent to which emergent behaviors depend on neural network function approximation remains an open research question. Without understanding the drivers of behavior, we cannot reliably predict when, how, or why exploration strategies will emerge, thereby limiting our ability to use these models safely and reliably.

Methodologically, our goal of *understanding* this phenomenon sits askew to the standard ML toolkit used to optimize performance on benchmark tasks. We thus take inspiration from cognitive science,

---

[*]Equal Contribution
Project page: https://mahsa-bastankhah.github.io/demystifying-single-goal-exploration/.

where researchers study intelligent behavior with a rich toolkit including rational analysis (Anderson, 1990), intervention experiments (Bower & Clapper, 1989), and cognitive modeling (McClelland, 2009). As a case study in how methods from cognitive science can be used to study the properties of AI models, we adapt these methods to understand emergent exploration in SGCRL. Specifically, we (1) theoretically analyze the optimization objective to uncover the implicit drivers of agent behavior, (2) conduct controlled intervention experiments on these behavioral drivers, and (3) build a simple model of the exploration mechanism in a tabular setting.

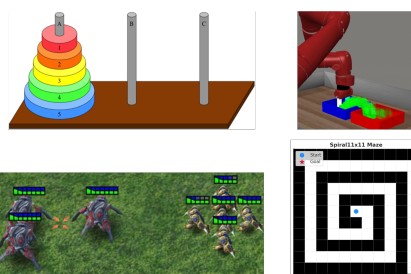

The main contribution of this paper is to show that it is possible to gain insight into the emergent behavior of even relatively complex learning algorithms. We do so by answering a specific question: *Why does the SGCRL algorithm explore effectively in the absence of any obvious intrinsic or extrinsic rewards?* We show that exploration is driven by the interplay between the actor and the critic. Although the algorithm is trained without external rewards, the actor's objective can be reinterpreted as **maximizing (implicit) rewards which drive the agent to states that look like the goal**, according to the agent's current representations. The critic shapes this implicit reward landscape by decreasing the representational goal similarity of states

Figure 1: SGCRL exhibits emergent exploration on a range of tasks. But why?

along unsuccessful trajectories, pruning them from future exploration. Surprisingly, a simplified tabular model of SGCRL reveals that these **exploration dynamics arise from learning low-rank representations, rather than from neural network function approximation**. Finally, our analysis provides insight into how to **adapt SGCRL to perform safety-aware exploration**. Our success in analyzing the emergent behavior of this particular algorithm suggests that this approach can be applied productively to other settings where AI systems produce complex and unexpected behaviors.

## 2 RELATED WORK

A growing line of work has discovered emergent behaviors in deep RL: agents demonstrate increasingly sophisticated skills without explicit programming (Liu et al., 2025; Bush et al., 2025; Simmons-Edler et al., 2025; Team et al., 2021). Existing approaches to understanding deep RL agents aim to improve algorithm transparency or explain behavior post-hoc (Heuillet et al., 2021; Glanois et al., 2024). Transparency-based methods include learning low-dimensional, meaningful representations of the state space (Lesort et al., 2018; 2019; Raffin et al., 2019), labeling actions based on targeted reward components (Juozapaitis et al., 2019), and maintaining subgoals with hierarchical RL (Beyret et al., 2019; Cideron et al., 2020). Although transparency-based methods improve algorithmic understanding, they require modified system architectures or auxiliary training tasks (Beyret et al., 2019; Juozapaitis et al., 2019). In contrast, post-hoc methods take the RL algorithm as a black-box and attempt to distill the representations or predictions into interpretable trees (Bewley & Lawry, 2021; Coppens et al., 2019) or maps (Greydanus et al., 2018; Zahavy et al., 2016). Our work differs from prior work in trying to explain behavior from an algorithmic perspective without the overhead of auxiliary training tasks.

Following a recent trend (Hamrick & Mohamed, 2020; Binz & Schulz, 2023; Frank, 2023; Ivanova, 2025; Ku et al., 2025), our methodology is inspired by tools in cognitive science for understanding intelligent behavior, including rational analysis, intervention experiments, and small-scale models. Rational analysis explains the behavior of agents by considering the optimal solutions to the problems that they face (Anderson, 1990). In the context of AI models, this means determining the consequences of maximizing their objective (McCoy et al., 2024). Intervention experiments are used to test theories about the causal mechanisms underlying behavior (Bower & Clapper, 1989). Building simple cognitive models is a tool that can be used to test whether a small set of principles is sufficient to reproduce that behavior (McClelland, 2009). Accordingly, we draw on rational analysis and controlled interventions to uncover the implicit objectives driving SGCRL's exploration dynamics and use a simplified computational model of the algorithm to understand how these dynamics arise.

Exploration in goal-conditioned RL is challenging because rewards are extremely sparse. Many prior methods address this by commanding data collection on a distribution of goals, even when the

downstream task involves only a single target, to encourage broad exploration during the training (Chane-Sane et al., 2021; Savinov et al., 2018; Shah et al., 2021; Zhang et al., 2021a; Florensa et al., 2018; Pong et al., 2019; Venkattaramanujam et al., 2019). However, Liu et al. (2025) showed that the contrastive RL algorithm, when collecting data with a *single hard goal*, induces strong exploratory behavior and allows the agent to acquire useful skills without additional supervision. They further demonstrated that this single-goal data collection approach outperforms methods based on a distribution of goals, goal curricula, and waypoint generation. In this work, we explain why single-goal training succeeds. Moreover, we show that collecting data with a single hard goal produces representations that drive more effective exploration than those learned from multi-goal data—demystifying the surprising result of Liu et al. (2025) that single-goal training can outperform subgoal curricula.

## 3 PRELIMINARIES

**Problem Setup.** We formulate a sequential decision-making problem as a Markov decision process (Bellman, 1957; Puterman, 1994) without an explicit reward function. The state space is denoted by $\mathcal{S}$ with the initial state is sampled $s_0 \sim p_0$ and subsequence states are sampled $s_{t+1} \sim p(\cdot \mid s_t, a_t)$. The agent is given a single target goal $g \in \mathcal{S}$, representing the desired state to be reached. The agent must learn a goal-conditioned policy $a_t \sim \pi(\cdot \mid s_t, g)$.

**Goal-conditioned RL.** For any policy $\pi$, we define the $\gamma$-discounted occupancy measure (Sutton et al., 1999a) $p_\gamma^\pi(s_f) := (1 - \gamma) \sum_{t=0}^{\infty} \gamma^t p_t^\pi(s_t = s_f)$, where $p_t^\pi(s_t = s_f)$ is the probability of being at state $s_f$ at timestep $t$. Following prior work (Blier et al., 2021; Chane-Sane et al., 2021; Eysenbach et al., 2022), the objective is to learn a policy that maximizes the probability of reaching the goal: $\max_\pi \ p_\gamma^\pi(g)$. This task is challenging because the agent does not receive any feedback from the environment regarding its intermediate progress towards success.

**Single-Goal Contrastive RL (SGCRL).** We study SGCRL (Liu et al., 2025), an actor–critic framework based on temporal contrastive learning. In contrast to prior work (Nasiriany et al., 2019; Chane-Sane et al., 2021), the exploration of SGCRL (Liu et al., 2025) is not guided by a human-designed curricula of tasks or manually specified reward functions. The critic estimates the likelihood that a state–action pair $(s, a)$ leads to a future state $s_f$, and is parameterized as $\phi(s, a)^\top \psi(s_f)$, where $\phi(s, a)$ and $\psi(s_f)$ are learned embeddings. The critic embeddings are trained with a contrastive loss where, for each state-action pair, $(s_t, a_t)$, positive future states are drawn by looking $\Delta \sim \text{Geom}(1 - \gamma)$ steps ahead; meanwhile, negative examples are sampled from the marginal distribution $p(s_f) := \mathbb{E}_{p(s,a)}[p_\gamma(s_f \mid s, a)]$. In particular we use the backward InfoNCE loss (Myers et al., 2024; Liu et al., 2025):

$$\max_{\phi, \psi} \ \mathbb{E}_{\substack{(s_i, a_i) \sim p_\mathcal{D}(s,a) \\ s_f^{(i)} \sim p_\gamma^\pi(\cdot \mid s_i, a_i) \\ i=1,\ldots,N}} \left[ \frac{1}{N} \sum_{i=1}^{N} \log \frac{\exp(\phi(s_i, a_i)^\top \psi(s_f^{(i)}))}{\sum_{j=1}^{N} \exp(\phi(s_j, a_j)^\top \psi(s_f^{(i)}))} \right], \tag{1}$$

where $p_\mathcal{D}(s, a)$ denotes the empirical data distribution of the replay buffer. The contrastive objective aligns each state-action pair with its true positive future state while discouraging alignments with unrelated states. We normalize all representations by their $\ell_2$ norm. Once trained, the critic encodes a log-$Q$ value $\phi(s, a)^\top \psi(s_f) = \log p_\gamma^\pi(s_f \mid s, a) - \log p(s_f)$, (Eysenbach et al., 2022).

The actor aims to select actions that maximize the likelihood of reaching the goal:

$$\max_{\pi(a \mid s, g)} \ \mathbb{E}_{s \sim p(s), \ a \sim \pi(\cdot \mid s, g)} \left[ \phi(s, a)^\top \psi(g) + \tau H(\pi(\cdot \mid s, g)) \right], \tag{2}$$

where $\tau$ is an entropy-regularization coefficient (Williams & Peng, 1991). In the discrete-action setting, the policy optimizing this objective samples actions from a softmax distribution:

$$\pi(a \mid s, g) = \frac{e^{\frac{1}{\tau} \phi(s,a)^\top \psi(g)}}{\sum_{a'} e^{\frac{1}{\tau} \phi(s,a')^\top \psi(g)}}. \tag{3}$$

For continuous action spaces, we train a parameterized actor $\pi(\cdot \mid s, s_f)$, where $s_f$ denotes a target state that is not restricted to the final hard goal. The algorithm always collects data conditioned on a single goal $g$ and does not make use of rewards, demonstrations, or subgoals. For a complete description of the method, see Appendix A.

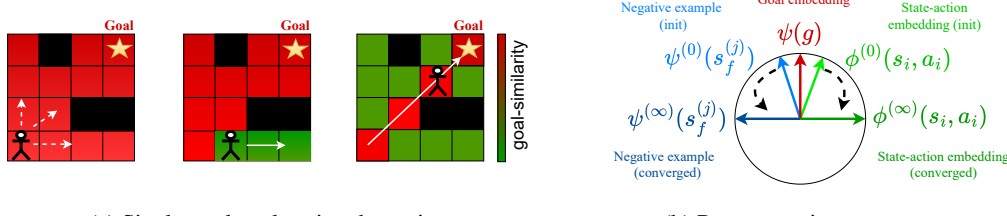

(a) Single-goal exploration dynamics          (b) Representation geometry

Figure 2: **(a)** The agent moves toward goal-like states (left), rules out non-goal states by assigning them low (green) $\psi$-similarity (center), and leaves a high (red) $\psi$-similarity trace along the successful trajectory once the goal is found (right). Solid lines indicate past paths; dashed lines indicate new paths. **(b)** Negative representations $\phi(s_i, a_i)$ and $\psi(s_f^{(j)})$ are pushed away from the common component $\psi(g)$, enhancing their contrast.

## 4  SGCRL REPRESENTATIONS INDUCE A CURRICULUM OF REWARDS

In this section, we theoretically characterize the dynamics that drive exploration in SGCRL and make testable hypotheses about its behavior, which we later verify in Section 5. We posit that the actor maximizes a discounted sum of implicit rewards shaped by the critic. Even when the algorithm poorly estimates the probability of reaching the goal, the implicit reward function is well-defined and directs exploration (Fig. 2a-left). As the actor optimizes this implicit reward signal, the critic dynamically reshapes the reward landscape. This implicit reward reflects the agent's belief about the goal position, which updates over time as the agent gathers more data. Before the goal is found, the critic decreases the implicit reward for states along unsuccessful trajectories (Fig. 2a-center). However, once the goal is reached, the critic instead increases rewards along the successful path (Fig. 2a-right), shifting the agent's behavior from exploration to exploitation. In this way, our analysis highlights a connection to a long line of work on reward shaping and the optimal design of reward functions that best facilitate efficient synthesis of good policy parameters (Ackley & Littman, 1992; Ng et al., 1999; Singh et al., 2009; Sorg et al., 2010a;b; Sorg, 2011; Devlin & Kudenko, 2012).

Moreover, since our analysis revolves exclusively around the associated actor and critic objectives, it would suggest that the exploratory behavior of SGCRL does not depend on function approximation.

Section 4.1 establishes the equivalence between SGCRL and a reward-maximizing agent that maximizes representation similarity. Section 4.2.1 shows how contrastive learning naturally reduces representation similarity in explored states, while Subsection 4.2.2 discusses how representations evolve once the goal is reached. Formal theorem statements and proofs are provided in Appendix B.

### 4.1  THE ACTOR MAXIMIZES AN IMPLICIT, REPRESENTATION-BASED REWARD

Our analysis reveals that, although the SGCRL objective is defined with respect to reaching $g$, it simultaneously drives the agent toward states $s_f$ that exhibit high representational similarity to the goal, quantified by the inner product between the state and goal representations, a metric we denote by $\psi$-similarity $:= \psi(s_f)^\top \psi(g)$. One interpretation of this similarity function is the agent's encoding of beliefs about the goal's location in the $\psi$-similarity metric, which is progressively refined through critic updates as more data are collected. This intuitively resembles posterior-sampling approaches, where the agent maintains a posterior distribution over the underlying reward and transition functions (Strens, 2000; Osband et al., 2013) to guide exploration via epistemic uncertainty (Der Kiureghian & Ditlevsen, 2009).

Our main result in this section relies on an alignment property of the InfoNCE objective (Eq. 1), namely that optimization encourages the representations of positive pairs to align in expectation. This assumption is consistent with insights from prior work (Wang & Isola, 2020). However, in Appendix B.2, we provide two concrete conditions under which the alignment property provably holds. For example, Appendix B.2.2 shows that alignment is guaranteed when the learned contrastive representations have an isotropic Gaussian distribution (an assumption corroborated by previous work

Wang & Isola (2020)). For clarity, we state the alignment property here as an assumption, with full proofs and technical details deferred to the Appendix.

**Assumption 1** (Alignment of positive examples under InfoNCE). *When the InfoNCE loss (Eq. 1) is optimized, the representation of any state–action pair $(s, a)$ aligns with the expected representation of its future states:*

$$\phi(s, a) \ = \ \mathbb{E}_{s_f \sim p_\gamma^\pi(s_f|s,a)}\big[\psi(s_f)\big].$$

**Theorem 1.** *Given Assumption 1, maximizing the likelihood of reaching the goal in the SGCRL actor objective (Eq. 2) is equivalent to maximizing the return*

$$\mathbb{E}_{\substack{s\sim p(s)\\a\sim\pi(\cdot|s,g)}}\Big[\phi(s,a)^\top\psi(g))\Big] \ = \ \mathbb{E}_{\substack{s\sim p(s)\\a\sim\pi(\cdot|s,g)}}\Big[Q^\psi(s,a)\Big],$$

*where*

$$Q^\psi(s,a) \ \coloneqq \ \mathbb{E}_{p_t^\pi(s_t)}\left[\sum_{t=0}^\infty \gamma^t\,\psi(s_t)^\top\psi(g) \ \Big| \ s_0 = s, \ a_0 = a\right].$$

Theorem 1 implies that, although SGCRL is an unsupervised algorithm and does not use any external reward from the environment, it implicitly maximizes an internal reward i.e., $\psi$-similarity. This result suggests that when $\psi$-similarity is well-structured and sufficiently dense, it can effectively drive exploration. For instance, initial high $\psi$-similarity near the initial state provides the agent with a gradient to move away from the start; however, $\psi$-similarity must decrease over time in regions without the goal, otherwise the agent risks becoming trapped. Moreover, Theorem 1 yields testable predictions, which we validate in Section 5. In particular, it predicts that the agent will be drawn toward regions of high $\psi$-similarity and avoid regions of low $\psi$-similarity, even when these regions are respectively far from or close to the goal (see Section 5.2 for the empirical validation). Moreover, Theorem 1 yields testable predictions, which we validate in Section 5. In particular, the theorem predicts that the agent's behavior is guided by similarity in *representation space*: the agent is attracted to states with high $\psi$-similarity to the goal embedding and repelled from states with low $\psi$-similarity, even when those states are respectively far from or close to the goal in the raw environment. We empirically confirm this prediction in Section 5.2.

Notably, the formulation in Theorem 1 resembles the structure of the successor representation (Dayan, 1993) and successor features (Barreto et al., 2017; Kulkarni et al., 2016), where $\psi(s)$ plays the role of a state feature vector and $\phi(s, a)$ resembles a successor feature prediction. Yet several conceptual differences distinguish our setting from this prior work. First, successor-feature methods explicitly train state features (e.g., via reconstruction losses) and then learn successor features on top of them. In SGCRL, both the features and their successor representations emerge naturally from the InfoNCE objective, without requiring explicit successor-feature training. Second, whereas successor features are typically used for rapid reward adaptation and transfer (Barreto et al., 2017; Kulkarni et al., 2016), our focus is on how the learned representations themselves drive exploration. In SGCRL, the state features evolve dynamically during training in a way that induces exploratory behavior. Third, although this shaping effect resembles potential-based reward shaping (Ng et al., 1999), SGCRL does not implement such shaping explicitly. The "potential" toward the single goal arises implicitly from the contrastive objective rather than from hand-crafted potentials. This contrasts with frameworks such as eigenoptions, which deliberately construct potential-based rewards for different options (Machado et al., 2017).

## 4.2 Representation Updates Advance the Implicit Reward Curriculum

So far, we discussed that the agent is driven by the shaped reward defined through $\psi$-similarity. In this section, we analyze the evolution of $\psi$-similarity as the representations are updated. Specifically, in Section 4.2.1 we show that before the goal is discovered, the $\psi$-similarity values of states that have already been explored gradually decrease, preventing the agent from revisiting them and thereby pruning the search space. In contrast, in Section 4.2.2 we argue that once the goal is found, this process reverses: states along the path to the goal acquire higher $\psi$-similarity after their representations are updated, which encourages the agent to consistently exploit that trajectory.

While the use of intrinsic rewards that exhibit this exploration-exploitation behavior has a rich history—including novelty bonuses, episodic or count-based bonuses, and prediction-error bonuses

(Mohamed & Jimenez Rezende, 2015; Bellemare et al., 2016; Burda et al., 2019; Raileanu & Rocktäschel, 2020; Pathak et al., 2017a; Henaff et al., 2022; Zhang et al., 2021b)—SGCRL differs fundamentally from these approaches. In prior methods, intrinsic rewards are heuristically designed, manually added to the task reward, and tuned via hyperparameters to balance exploratory drive with task-directed performance and exploitation. By contrast, the intrinsic reward in SGCRL is not an external design choice but emerges directly from the actor objective; it is principled, requires no additional tuning, and comes with an intuitive theoretical guarantee: it corresponds exactly to maximizing the probability of reaching the goal under the agent's current knowledge about candidate goal states. In contrast, while other principled mechanisms for incentivizing exploration exist!(Agarwal et al., 2020a;b), empirical demonstrations of their practicality and scalability remain limited compared to SGCRL.

### 4.2.1 EXPLORATION BEFORE GOAL DISCOVERY

We analyze a simplified setting in which $\psi(g)$ is fixed and the agent explores a region of the environment that does not contain the goal; this assumption is realistic when the states are sufficiently far from the goal and the critic network is sufficiently expressive. In this case, the updates affect only the representations of states in that triplet, i.e., $(s, a, s_f)$, while leaving $\psi(g)$ unchanged. The following theorem shows that if states in a region initially exhibit consistently high $\psi$-similarity (that is, if their representations share a common component parallel to the goal representation, in addition to independent state-specific noise) then repeated InfoNCE updates will reduce their similarity to $\psi(g)$ until they become orthogonal, rendering the region unattractive to the actor. Intuitively, since the InfoNCE loss is invariant to shared components (i.e., adding a fixed vector to all representations) the component of the representations parallel to $\psi(g)$ does not help learn temporal differences. Because normalized representations have limited capacity, they suppress this redundant component to better use their representational budget to minimize the loss. See Figure 2b for an intuitive illustration.

**Theorem 2** (Informal). *Let $\mathcal{D} = \{(s_i, a_i, s_f^{(i)})\}_{i=1}^N$ denote the collected dataset, where each triplet consists of a state $s_i$, an action $a_i$, and the corresponding future state $s_f^{(i)}$ observed in the trajectory after taking $(s_i, a_i)$. Let $\psi(g) \in \mathbb{R}^d$ be a fixed high-dimensional unit vector representing the goal such that $g \notin \{s_f^{(i)}\}_{i=1}^N$. Consider the normalized anchor embeddings $\{\phi(s_i, a_i)\}_{i=1}^N$ and future embeddings $\{\psi(s_f^{(i)})\}_{i=1}^N$ that are initialized as follows:*

$$\phi^{(0)}(s_i, a_i) = c\,\psi(g) + \zeta_i, \qquad \psi^{(0)}(s_{f,i}) = c\,\psi(g) + \kappa_i,$$

*where $\zeta_i, \kappa_i \in \mathbb{R}^d$ are i.i.d isotropic Gaussian vectors and $c \in \mathbb{R}$ is a non-zero scalar. Suppose these embeddings are updated using the InfoNCE gradient descent update rule as specified in Appendix B.1, with a sufficiently large batch size $N$ and sufficiently small learning rate $\eta$. Then, with high probability over the random initialization, the system converges to an equilibrium that satisfies $\phi(s_i, a_i)^\top \psi(g) = \psi(s_f^{(i)})^\top \psi(g) = 0, \forall i$.*

Notably, the proof of this theorem relies only on a fixed-point analysis of the InfoNCE loss and does not require any neural network function approximation.

In practice, at initialization, all $\psi$ representations are nearly the same (no temporal structure has been learned), so $c$ is high for all states. As training progresses, states along unsuccessful trajectories move away from $\psi(g)$ in representation space. Consequently, Theorems 1 and 2 reveal a two-player dynamic: the actor seeks regions with high $\psi$-similarity, while the critic reduces their similarity when the goal is absent. Refer to Appendix B.5 for a discussion on how this mechanism enables efficient exploration even in continuous settings, where the set of states to be ruled out is infinite. Theorem 2 assumes that $\psi(g)$ remains fixed while other representations are updated—an assumption that holds exactly in the tabular setting. With a shared neural encoder, updates to distant states can slightly shift $\psi(g)$, but the theorem is stable to such drift: if $\psi(g)$ changes by at most $\varepsilon$, orthogonality holds up to an $\varepsilon$ error. Moreover, we show in Appendix D.11, that this orthogonalization effect persists empirically in continuous settings, indicating that the fixed-point analysis remains a good practical approximation.

### 4.2.2 EXPLOITATION AFTER GOAL DISCOVERY

Considering that contrastive learning aligns the representations of positive examples (See Assumption 1, Appendix B.2), the representations along a successful trajectory to the goal should align with the goal representation $\psi(g)$, since $g$ appears as a positive example for those states. This leaves a "trace" of high $\psi$-similarity states that enables the agent to reliably rediscover the goal. While full alignment cannot be guaranteed theoretically, we find empirically that successful trajectories indeed form a high $\psi$-similarity trace from the start state to the goal. As shown in Section 5.1, this trace consistently guides the agent to the goal, marking the transition from exploration to exploitation once the goal has been discovered.

## 5 EXPERIMENTS

In this section, we present empirical evidence supporting our theoretical characterization of single-goal exploration (Section 4) in both continuous and tabular settings. Through our experiments, we address the following research questions:

**RQ1.** How do critic representations evolve during training to facilitate exploration and exploitation?

**RQ2.** How do critic representations influence the actor's data collection strategy?

Subsection 5.1 addresses RQ1 using both a simplified model of SGCRL in a tabular setting as well as standard SGCRL in the continuous setting. Subsection 5.2 addresses RQ2 in the continuous setting and motivates our findings with preliminary results on how to utilize goal-similarity to improve safety.

**Tasks.** We study SGCRL on 2D point maze navigation tasks adapted from prior work (Eysenbach et al., 2022; Liu et al., 2025) as well as the Tower of Hanoi goal-reaching task. The navigation tasks involve reaching a goal in various maze configurations including the classic Four Rooms domain (Sutton et al., 1999b), an L-shaped wall, and a spiral wall. The Tower of Hanoi task involves moving a stack of disks across three locations with the constraint that larger disks cannot be placed on smaller ones, and has been used extensively in studies of human problem-solving in cognitive science (Simon, 1975). We do not use rewards or subgoals to solve any of the tasks, and success is determined by whether the goal state is reached at some point during an episode. All training metric curves are averaged over 8 random seeds. All shading denotes one standard error.

**Tabular SGCRL.** Previous implementations of SGCRL utilize neural network function approximation, raising the question of whether these behaviors are fundamental properties of the SGCRL algorithm or artifacts of neural network dynamics. To isolate the SGCRL exploration mechanism from neural network generalization properties, we designed a simplified computational model of the algorithm for the tabular FourRooms maze and Tower of Hanoi task. Each state $s$ has an embedding $\psi(s)$ stored in a lookup table and updated via the InfoNCE gradient rule (Appendix B.1). We assume the environment follows deterministic transition dynamics and, instead of learning $\phi(s, a)$, further assume access to the ground-truth dynamics $s_{t+1} = p(s_t, a_t)$ (this latter assumption can be relaxed by learning in a model-based fashion). The policy takes actions according to Equation 3. Following the assumptions for Theorem 2, all representations are initialized as $\psi(s) = \mathbf{x} + \boldsymbol{\varepsilon}(s)$, with a global Gaussian seed $\mathbf{x}$ shared across states and small, independent Gaussian noise $\boldsymbol{\varepsilon}(s)$ per state.

### 5.1 RQ1. HOW DO CRITIC REPRESENTATIONS EVOLVE TO FACILITATE EXPLORATION AND EXPLOITATION?

Our theory suggests that SGCRL automatically develops a curriculum of subgoals by updating representations such that unsuccessful paths become less appealing over time (Thm. 2). To observe how representations change in a controlled setting, we conducted a series of experiments with both the simplified tabular SGCRL model and standard SGCRL algorithm, showing that $\psi$-similarity decreases for states along unsuccessful trajectories and increases for states along successful trajectories. These results support the hypothesis that SGCRL benefits from a natural exploration curriculum that progressively pushes the agent toward unexplored regions.

**Distinct phases of representation updates emerge.** By running a simplified tabular version of SGCRL, we aim to test whether SGCRL's mechanism arise primarily from contrastive learning

of low-rank representations rather than from generalization properties of neural networks. Our characterization of SGCRL's exploration mechanism prescribes that representation updates should be distinct before finding the goal compared to after finding the goal. That is, before reaching the goal, we would expect the $\psi$-similarity of frequently visited states to decrease, and after finding the goal, we would expect the $\psi$-similarity of frequently visited states to increase. To verify these dynamics, we ran tabular SGCRL in the Tower of Hanoi environment. We measured the correlation between state visitation and goal similarity before and after the agent reached the goal.

We find that, even with the simplified tabular model, the agent demonstrates effective exploration and goal-reaching with distinct phases of behavior. Given uniform initialization of all states close to the goal, the correlation between state visitation and goal similarity starts high. As training progresses, the state visitation count and goal similarity become negatively correlated prior to reaching the goal and positively correlated after reaching the goal (see Fig. 3). We also performed an ablation study where we replaced the vectorized representations with a $|\mathcal{S}| \times |\mathcal{S}|$ lookup table of state-goal similarity and updated these scalar

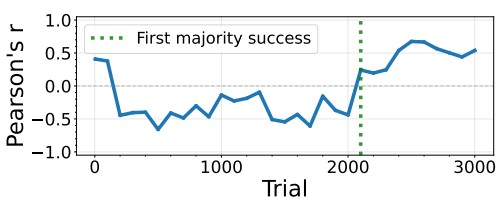

Figure 3: Running SGCRL in a tabular Tower of Hanoi environment reveals distinct phases of learning. The 3 disk task is shown, but results generalize to 4/5 disks.

values with the same contrastive objective. In this setting, the agent fails to explore efficiently, requiring $\sim 100$x more samples than tabular SGCRL. (Appendix D.3). These results indicate that SGCRL's exploration dynamics arise from contrastive learning with low-rank representations rather than from neural network approximation or from the contrastive learning objective alone.

In Appendix D.1, we compare the evolution of $\psi$-similarity in SGCRL with the value function of R-MAX (Brafman & Tennenholtz, 2002), an optimism-based exploration method for the tabular setting that is provably-efficient (Strehl et al., 2009). R-MAX and SGCRL share similar exploration dynamics: R-MAX starts with the belief that all states yield the maximum reward (equivalent to assigning high $\psi$-similarity in SGCRL), and exploration progressively corrects these estimates as states are visited and their true rewards are revealed. We also refer the reader to Appendix D.6, which shows that although the exploration dynamics of SGCRL could, in the very worst case, require searching the entire state space to reduce all $\psi$-similarity values, this does not occur in practice. In continuous settings represented by neural networks, the algorithm avoids exhaustive search. Although our findings suggest that neural network generalization is not the primary reason for SGCRL's exploration mechanism, it nevertheless provides a useful inductive bias: $\psi$-similarity reductions for unsuccessful paths generalize to nearby states, improving exploration efficiency.

**Representations along unsuccessful trajectories become dissimilar to the goal.** To study RQ1 further, we simulate how representations evolve when goals are unreachable. Theorem 2 predicts that representations for states along unsuccessful trajectories should become orthogonal to the goal. To test this prediction, we ran an experiment in which we assigned the agent an imaginary goal $\psi(g) = \mathbf{z}$, with $\mathbf{z}$ sampled from a Gaussian distribution. We projected the learned representations into three dimensions using PCA (see Fig 4), with $\mathbf{z}$ aligned to the vertical axis. We find that initially the representations cluster near the goal at the top of the unit sphere. Over time, they drift toward the "equator", collapsing into the subspace orthogonal to $\mathbf{z}$, while still clustering states from the same

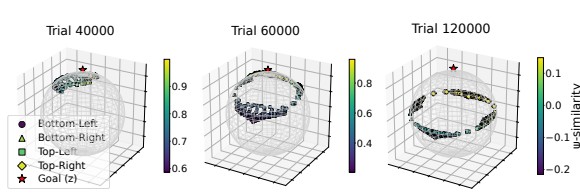

(a) Representation evolution

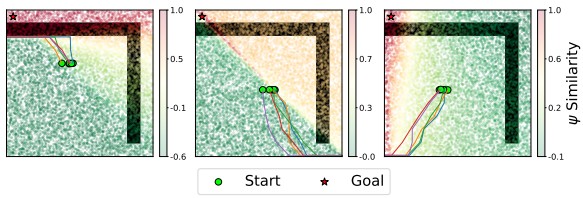

(b) $\psi$-similarity evolution

Figure 4: **(a)** Initially all representations start close to $z$, later collapsing into a subspace orthogonal to $z$, while preserving local room-level structure. **(b)** $\psi$-similarity decreases along traversed, unsuccessful paths.

room together. The drift reflects the agent's visitation pattern: states in the bottom-left room, visited first, are pulled away earliest, while states in the top-right room, reached only in later episodes, collapse last. This experiment highlights a key property of the SGCRL data collection strategy: Because the single-goal conditioned policy consistently drives the agent toward areas with higher $\psi$-similarity (unvisited states), representations of previously visited states are continually pushed further from the goal. These results provide insight into RQ1 and highlight the effectiveness of single-goal exploration, even in settings without a real goal. Moreover, the same exploration dynamic also extends to the multi-goal setting. With a simple modification to the action-selection mechanism, SGCRL is able to reach a distribution of goals (Appendix D.7).

To test whether the same representation trend holds for the continuous setting, we conduct an experiment using standard SGCRL in which we fixed the data collection trajectories to always move in a particular cardinal direction. This experiment allows us to isolate the effect of unsuccessful visitation on representational similarity. We find that initially the $\psi$-similarity of all states is high, but, over the course of training, states along the frequently traversed paths systematically become less similar to the goal (see Fig 4b).

**SGCRL data collection structures representations strategically.** We also investigate whether the single-goal data collection strategy offers distinct advantages over other data collection strategies for the representation shaping described in the above experiment. For the FourRooms task, we compare single-goal data collection with an alternative strategy that samples goals from a uniform distribution. This strategy fails to push representations of frequently visited states far from the goal, preventing effective exploration (Appendix D.5). These results indicate that the SGCRL actor is not merely a consumer of well-formed representations, but also an active contributor. Through data collection, it shapes representations in a principled way that decreases $\psi$-similarity for frequently visited states.

## 5.2 RQ2. How do critic representations influence the actor's data collection strategy?

Next, we investigate how the dynamic representation updates characterized in Section 5.1 influence the actor's data collection strategy. Our theory posits that goal-similarity provides an internal reward (Thm. 1) to direct agent behavior. In this vein, we conduct intervention experiments to study how goal-similarity influences SGCRL's visitation behavior, with applications to safety-aware exploration.

**Agent targets states with high $\psi$-similarity.** In our first experiment, we perturb the initial position of the agent at various training checkpoints to be closer to the goal and observe whether the agent directly targets the goal state or target states that "look like" the goal (high $\psi$-similarity). We find that the agent navigates toward the closest region with high representational goal similarity, even if it can reach the goal directly through a shorter path (see Fig. 5a). To test this behavior further, we conduct another intervention in which we fixed the representations of a patch in the top-right room of the FourRooms environment to match the goal embedding $\psi(g)$. As shown in Fig. 5b (top), the agent is strongly attracted to this patch, leading to a substantial increase in visitation of the top-right room compared to the control setup (Fig. 5b, bottom). Notably, even when the agent succeeds in reaching the true goal, it frequently detours into this patch along the way. These results yield an answer to RQ2, indicating that the agent is guided by an implicit reward signal based on representational similarity to the goal. This behavior emerges from the learning objective without any explicit programming to seek goal-like states.

**Agent avoids states with low $\psi$-similarity.** Based on our characterization of SGCRL's behavioral drivers (Sec. 4), we predict that manipulating contrastive representations allows more fine-grained control of agent behavior during both training and deployment. We tested this prediction by setting the representations of the states in one of the rooms in the continuous FourRooms environment to be the negative of the goal representation ($\psi(s) = -\psi(g); \forall s \in \mathcal{R}$). We find that this intervention leads the agent to systematically avoid that region during both training (Figure 5c – top) and test time (Figure 5c – bottom). The agent successfully finds alternate paths to the goal while respecting the imposed constraints (see Appendix D.15 for more figures). These preliminary experiments show that understanding goal representation-driven exploration could lead to improvements in safety and control of goal-reaching tasks, enabling practitioners to guide agent behavior through representation design rather than explicit reward engineering (Ibrahim et al., 2024).

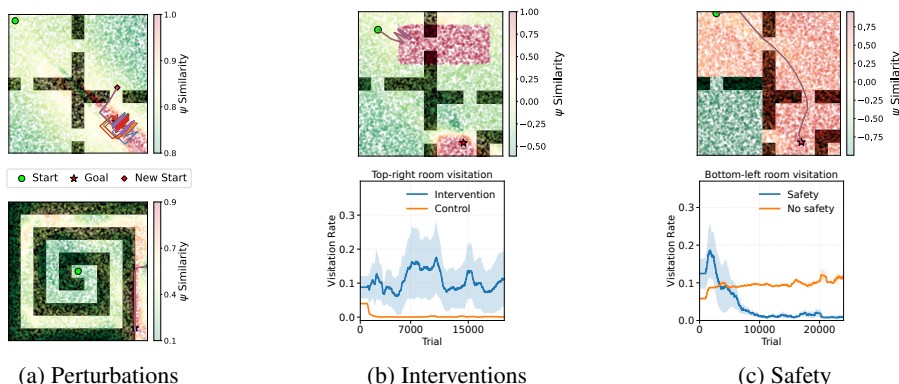

(a) Perturbations $\qquad$ (b) Interventions $\qquad$ (c) Safety

Figure 5: SGCRL targets states with high $\psi$-similarity **(a, b)**, avoiding states with low $\psi$-similarity **(c)**

# 6 CONCLUSION

Through theoretical analysis and controlled experiments, we have shown that SGCRL implicitly maximizes rewards based on representational goal-similarity, enabling effective exploration without explicit rewards. Our results indicate that these exploration dynamics arise from contrastive learning of low-rank representations rather than from function approximation with neural networks.

Prior work in self-supervised learning has considered contrastively-learned low-rank representations detrimental to downstream classification tasks due to loss of representational capacity. However, for SGCRL, limited representational capacity is not a weakness, but rather a necessary component of the method's success. More specifically, the mechanism that allows SGCRL to prune the search space relies on the geometric constraints imposed by low-dimensional, normalized embeddings. Before the goal is found, the InfoNCE objective suppresses shared components (e.g. psi(g)) that do not help learn temporal differences, driving the representations of states along unsuccessful trajectories to become orthogonal to the goal embedding and incentivizing exploration towards unvisited states. At a high-level, these dynamics align with classic exploration algorithms — like R-MAX and PSRL — that refine a set of candidate desirable states during exploration. This characterization not only helps to explain the success of SGCRL in prior work, but also charts a path for how to retain the strong, appealing theoretical properties of R-MAX/PSRL in high-dimensional as well as long-horizon tasks (Liu et al., 2025; Eysenbach et al., 2022; Zheng et al., 2023).

Beyond analyzing SGCRL specifically, our analysis provides a case study for understanding emergent exploration through the lens of algorithmic interpretability inspired by cognitive-science methods. We draw on analysis techniques commonly used in cognitive science—including rational analysis, controlled interventions, and simplified modeling, to construct and test a theoretical account of SGCRL's behavior. The particular instantiation of this framework operationalized in this work successfully yields insight into the behavioral drivers of exploration, enabling us to better control these systems for safer deployment. We anticipate that a similar methodology can be used to gain deeper insight into a range of RL algorithms, potentially identifying ways in which those algorithms can be improved through the same implicit, contrastive reward-shaping process (Asmuth et al., 2008).

## 6.1 LIMITATIONS

Our empirical study, through its focus on SGCRL, exclusively revolved around goal-reaching tasks and did not consider reward maximization more broadly. We analyzed the learning dynamics of SGCRL but have yet to establish formal theoretical guarantees on its sample efficiency. In future work, we aim to study whether SGCRL can achieve polynomial sample complexity in the tabular setting (Kakade, 2003; Strehl et al., 2009) and extend the method empirically to a broader space of tasks beyond goal reaching.

## REPRODUCIBILITY STATEMENT

We have provided the source code for tabular SGCRL here. The experiments using standard SGCRL use the codebase and default parameters given in (Liu et al., 2025). The hyperparameters used for both tabular and standard SGCRL are given in Appendix C. The proofs of all theoretical results are provided in Appendix B.

## ETHICS STATEMENT

Our work investigates the exploration dynamics of a self-supervised reinforcement learning algorithm and therefore has no immediate ethical concerns. We also develop a variant that mitigates certain behaviors relevant to safety-critical applications. However, as with many advances in RL, similar techniques could, in principle, be misused to enhance adversarial behavior.

## ACKNOWLEDGMENTS

Thanks to the members of the Princeton RL lab for feedback on preliminary versions of the work. DA and TG were supported by ONR MURI N00014-24-1-2748 and ONR grant N00014-23-1-2510. GL acknowledges support by the National Science Foundation Graduate Research Fellowship under Grant No. DGE2140739. BE and MB acknowledge support from the National Science Foundation under Award No. 2441665. Any opinions, findings and conclusions or recommendations expressed in this material are those of the author(s) and do not necessarily reflect the views of the National Science Foundation.

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

APPENDIX

## A SINGLE-GOAL CONTRASTIVE RL ALGORITHM

In this section, we provide more details about the SGCRL algorithm as presented in (Liu et al., 2025).

---

**Algorithm 1 Single-goal Exploration with Contrastive RL.** The difference from most prior methods is that exploration is done by commanding a single difficult goal $s^*$, rather than sampling goals with a range of difficulties.

---

1: Initialize policy $\pi_\theta(a \mid s, g)$, replay buffer $\mathcal{B}$, classifier with logits $\phi(s, a)^T \psi(s_f)$.
2: **while** not converged **do**
3:     Collect one trajectory of experience using $\pi(a \mid s, s_f = s^*)$, add to buffer $\mathcal{B}$.
4:     Update representations $\phi(s, a), \psi(s_f)$ and policy $\pi(a \mid s, s_f)$ using contrastive RL.
5: Return policy $\pi(a \mid s, g = s^*)$.

---

SGCRL is a simple modification of contrastive RL (Eysenbach et al., 2022): rather than asking the human user to provide training subgoals for exploration, SGCRL always commands the policy to collect data with a single hard goal $s^*$. This single hard goal is chosen to be a semantically meaningful state corresponding to task completion. The actor and critic objectives are presented in Section 3.

## B THEORETICAL RESULTS

### B.1 GRADIENT DESCENT UPDATES FOR OPTIMIZING THE INFONCE OBJECTIVE

We derive the gradients of the backward InfoNCE loss with respect to the different representation parameters in order to characterize their update dynamics. Consider the batch $D = \{s_i, a_i, sf_i\}_N$ where $sf_i$ is the positive example for state action pair $s_i, a_i$.

$$\mathcal{L}(D; \phi, \psi) = -\sum_i \log \frac{\exp\left(\phi(s_i, a_i)^\top \psi(sf_i)\right)}{\sum_k \exp\left(\phi(s_k, a_k)^\top \psi(sf_i)\right)}$$

$$p_{ij} := \frac{\exp\left(\phi(s_i)^\top \psi(sf_j)\right)}{\sum_k \exp\left(\phi(s_k, a_k)^\top \psi(sf_j)\right)} \tag{4}$$

Note that $\sum_i p_{ij} = 1$ but $\sum_j p_{ij} \neq 1$

$$\nabla_{\phi(s_i, a_i)} \mathcal{L} = -\sum_j (\delta_{i,j} - p_{ij}) \cdot \psi(sf_j) \tag{5}$$

$$\nabla_{\psi(sf_j)} \mathcal{L} = -\sum_i (\delta_{i,j} - p_{ij}) \cdot \phi(s_i, a_i) \tag{6}$$

$$\phi^{(t)}(s_i, a_i) = \phi^{(t-1)}(s_i, a_i) + \eta \sum_j (\delta_{i,j} - p_{ij}) \cdot \psi^{(t-1)}(sf_j) \tag{7}$$

$$\psi^{(t)}(sf_j) = \psi^{(t-1)}(sf_j) + \eta \sum_i (\delta_{i,j} - p_{ij}) \cdot \phi^{(t-1)}(s_i, a_i) \tag{8}$$

where $\eta$ is the learning rate and $\delta_{i,j} := \mathbf{1}[i = j]$. The update rules for the forward loss are analogous; the only difference is that, in the denominator of $p_{ij}$, the summation is taken over $\psi(sf_k)$.

When representations are meant to have unit norm, we perform an additional normalization step after every update.

### B.1.1 CONVERGENCE.

**Definition 1** (Convergence). *The convergence of the system characterized by Equations 7 and 8 is a configuration in which the updates cease to alter any of the representations.*

*In the* non-normalized *case, this corresponds to vanishing gradients:*

$$\nabla_{\phi(s_i,a_i)}L \ = \ \nabla_{\psi(sf_i)}L \ = \ 0 \quad \forall i.$$

*In the* normalized *case, convergence arises either when the gradients vanish, or when they are parallel to the representations themselves, i.e.,*

$$\nabla_{\phi(s_i,a_i)}L \ = \ c_i' \, \phi(s_i, a_i), \qquad \nabla_{\psi(sf_i)}L \ = \ c_i \, \psi(sf_i), \quad \forall i,$$

*for some scalars $c_i, c_i'$. In this case, normalization preserves the representation since scaling by a constant leaves the direction unchanged:*

$$\frac{(c_i'+1)\phi(s_i, a_i)}{\|(c_i'+1)\phi(s_i, a_i)\|} \ = \ \phi(s_i, a_i),$$

*and similarly for $\psi(sf_i)$.*

## B.2 SUPPORTING ASSUMPTION 1

Here we provide two illustrative cases that support Assumption 1.

The first scenario considers the setting where representations are normalized, high-dimensional, and—at convergence—uniformly distributed on the unit sphere. The uniformity assumption is a well-established fact in the literature, with prior work showing that the optimum of contrastive losses indeed yields such distributions (Wang & Isola, 2020).

The second scenario addresses the case where the data trajectories used for training can be partitioned into disjoint trajectories. In this setting, we show that the $\phi$ and $\psi$ representations along each trajectory fully align—not only in expectation, but exactly—for all state–action pairs and future states sampled along that path. This provides an even stronger version of Assumption 1.

We formalize the first scenario in Lemma 1, and then establish the second case in Lemma 2.

### B.2.1 ALIGNMENT OF POSITIVE EXAMPLES UNDER DISJOINT TRAJECTORY ASSUMPTION

Consider the following data collection assumption: we collect data consisting of one anchor pair $(s_i, a_i)$ together with multiple future states $\{s_{f,i}^k\}_{k=1}^K$ for each trajectory. The trajectories are disjoint, meaning that no anchor pair $(s_i, a_i)$ or future state $s_{f,i}^k$ is shared across different trajectories. Under this assumption, we can state the following lemma (the proof of the lemma is easily extendable to cases where there are multiple anchor per trajectory):

**Lemma 1** (Positive examples have fully aligned representations). *Let $N \gg 1$ be the batch size and $d \gg 1$ be the representation dimension. Consider anchors $(s_i, a_i)$ and their corresponding positive states $sf_i$, $i = 1, \ldots, N$, with initializations*

$$\phi^0(s_i, a_i) = \zeta_i, \quad \psi^0(sf_i) = \kappa_i,$$

*where $\zeta_i^0, \kappa_i^0 \overset{i.i.d.}{\sim} \mathcal{N}\big(0, \frac{1}{d}I_d\big)$. Suppose the representations are updated using gradient descent update rule of either the backward or forward InfoNCE loss (with normalization), with update rules as characterized in Appendix B.1. Then, with high probability over the initialization, the following holds at equilibrium:*

$$\phi(s_i, a_i) = \psi(sf_i), \quad \forall i.$$

*Moreover, if each anchor $(s_i, a_i)$ has more than one positive example $\{sf_i^k\}_k$, then at equilibrium*

$$\phi(s_i, a_i) = \psi(sf_i^k), \quad \forall i, k,$$

*where $k$ indexes the positive examples.*

*Proof.* We refer the reader to the proof of Theorem 3 for the first part of this proof. There, we analyze InfoNCE convergence under the assumption that each anchor $(s_i, a_i)$ has only a single future state $s_i^f$. In particular, Claim 2 of Theorem 3 establishes the result in this case.

The more realistic setting, however, is when each anchor is associated with multiple positive examples. To handle this, we slightly adapt the proof of Theorem 3 to again establish full alignment. We use the same notation as in that proof and restrict attention to the case where each anchor has $K = 2$ positive examples. The extension to any $K > 1$ follows identically.

Concretely, we assume a batch of size $N$, with anchor representations denoted $\{\mathbf{u}_i\}_{i=1}^N$ (corresponding to $\phi(s_i, a_i)$) and positive representations $\{\mathbf{v}_i\}_{i=1}^N$ (corresponding to $\psi(sf_i)$). Each anchor $\mathbf{u}_i$ is duplicated, i.e., $\mathbf{u}_{2i} = \mathbf{u}_{2i+1}$, while its two positive examples $\mathbf{v}_{2i}, \mathbf{v}_{2i+1}$ are independent. In other words, each $\mathbf{u}_{2i}$ has two distinct positive examples.

We establish the following claims by induction, grouping indices $(2i, 2i + 1)$ into bundles:

(a) $\alpha^{(t)} := \langle \mathbf{v}_{2i}^{(t)}, \mathbf{u}_{2i}^{(t)} \rangle = \langle \mathbf{v}_{2i+1}^{(t)}, \mathbf{u}_{2i}^{(t)} \rangle$ does not depend on the choice of $i$, and $\alpha^t > 0, \forall t \geq 1$, and $\alpha^{(\infty)} = 1$.

(b) $\beta^{(t)} := \langle \mathbf{v}_{2i}^{(t)}, \mathbf{v}_{2i+1}^{(t)} \rangle$ does not depend on the choice of $i$, and $\beta^t > 0, \forall t \geq 2$, and $\beta^{(\infty)} = 1$.

(c) $\lambda^{(t)} := \langle \mathbf{v}_i^{(t)}, \mathbf{v}_j^{(t)} \rangle = \langle \mathbf{u}_i^{(t)}, \mathbf{v}_j^{(t)} \rangle = \langle \mathbf{u}_i^{(t)}, \mathbf{u}_j^{(t)} \rangle = 0$ whenever $\lfloor i/2 \rfloor \neq \lfloor j/2 \rfloor$; that is, cross-inner products are zero across bundles.

These properties hold at initialization ($t = 0$), since independent Gaussian vectors are almost surely orthogonal in high dimensions (see, for instance, Equation 3.14 of Vershynin (2018)). We show that the update dynamics then preserve these index invariant properties We simplify the probability matrix as defined in Equation 4 using the induction assumptions at step $t - 1$:

$$p := \frac{\exp \langle \mathbf{u}_i^{(t-1)}, \mathbf{v}_j^{(t-1)} \rangle}{\sum_k \exp \langle \mathbf{u}_k^{(t-1)}, \mathbf{v}_j^{(t-1)} \rangle}, \quad \text{if } \lfloor i/2 \rfloor = \lfloor j/2 \rfloor,$$

and

$$q := \frac{\exp \langle \mathbf{u}_i^{(t-1)}, \mathbf{v}_j^{(t-1)} \rangle}{\sum_k \exp \langle \mathbf{u}_k^{(t-1)}, \mathbf{v}_j^{(t-1)} \rangle}, \quad \text{if } \lfloor i/2 \rfloor \neq \lfloor j/2 \rfloor.$$

These satisfy the normalization condition $2p + (N - 2)q = 1$ and, since $N$ is large, $q$ is small.

Now we write the GD update rule noting that each anchor $\mathbf{u}_i$ receives two gradient updates per iteration (corresponding to its two positives):

$$\hat{\mathbf{u}}_{2i+1}^{(t)} = \hat{\mathbf{u}}_{2i}^{(t)}$$

$$= \mathbf{u}_{2i}^{(t-1)} + \eta \left( (1 - 2p) \left( \mathbf{v}_{2i}^{(t-1)} + \mathbf{v}_{2i+1}^{(t-1)} \right) - 2 \sum_{j \neq 2i, 2i+1} q \, \mathbf{v}_j^{(t-1)} \right),$$

$$\mathbf{u}_{2i}^{(t)} = \frac{\hat{\mathbf{u}}_{2i}^{(t)}}{\|\hat{\mathbf{u}}_{2i}^{(t)}\|}, \qquad \mathbf{u}_{2i+1}^{(t)} = \frac{\hat{\mathbf{u}}_{2i+1}^{(t)}}{\|\hat{\mathbf{u}}_{2i+1}^{(t)}\|},$$

$$\hat{\mathbf{v}}_{2i}^{(t)} = \mathbf{v}_{2i}^{(t-1)} + \eta \left( (1 - 2p) \, \mathbf{u}_{2i}^{(t-1)} - \sum_{j \neq 2i, 2i+1} q \, \mathbf{u}_j^{(t-1)} \right),$$

$$\mathbf{v}_{2i}^{(t)} = \frac{\hat{\mathbf{v}}_{2i}^{(t)}}{\|\hat{\mathbf{v}}_{2i}^{(t)}\|}.$$

We first prove (c). For simplicity, we denote $\alpha^{(t-1)} = \alpha$ and $\beta^{(t-1)} = \beta$. We also note that by the induction hypothesis, the norms $\|\hat{\mathbf{u}}_i^{(t)}\|^2$ and $\|\hat{\mathbf{v}}_i^{(t)}\|^2$ are independent of the index $i$ and we denote

these norms at time $t-1$ by $r_u$, $r_v$. We write the cross inner product and simplify it using the fact that $\lambda^{(t-1)} = 0$, $q \approx 0$ and $\eta$ is small and representations at time $t-1$ are unit norm.

$$
\begin{aligned}
\langle \mathbf{u}_i^{(t)}, \mathbf{v}_j^{(t)} \rangle &= \frac{\langle \hat{\mathbf{u}}_i^{(t)}, \hat{\mathbf{v}}_j^{(t)} \rangle}{\|\hat{\mathbf{u}}_i^{(t)}\| \|\hat{\mathbf{v}}_i^{(t)}\|} \\
&= \frac{1}{\sqrt{r_u r_v}} \Big[ -\eta q (3 + \beta) + \mathcal{O}(\eta^2) \Big] \underset{q \approx 0}{\approx} 0, \quad \text{whenever } \lfloor i/2 \rfloor \neq \lfloor j/2 \rfloor.
\end{aligned}
$$

The proofs for the cross inner products $\langle \mathbf{v}_i^{(t)}, \mathbf{v}_j^{(t)} \rangle$ and $\langle \mathbf{u}_i^{(t)}, \mathbf{u}_j^{(t)} \rangle$ when $\lfloor i/2 \rfloor \neq \lfloor j/2 \rfloor$ are entirely analogous to the argument above. Note that the invariance to the index, carry from timestep $t-1$ to $t$. Now we analyze the evolution of $\beta^{(t)}$ and $\alpha^{(t)}$, which similarly remain invariant to the choice of index at time $t$, if being invariant to index at time $t-1$.

$$
\begin{aligned}
r_u^{(t)} &:= \left\| \mathbf{u}_{2i}^{(t-1)} + \eta \left( (1 - 2p)(\mathbf{v}_{2i}^{(t-1)} + \mathbf{v}_{2i+1}^{(t-1)}) - 2 \sum_{j \neq 2i, 2i+1} q \, \mathbf{v}_j^{(t-1)} \right) \right\|^2 \\
&= 1 + \mathcal{O}(\eta^2) + 4\eta(1 - 2p)\alpha, \tag{9}
\end{aligned}
$$

$$
\begin{aligned}
r_v^{(t)} &:= \left\| \mathbf{v}_{2i}^{(t-1)} + \eta \left( (1 - 2p) \mathbf{u}_{2i}^{(t-1)} - \sum_{j \neq 2i, 2i+1} q \, \mathbf{u}_j^{(t-1)} \right) \right\|^2 \\
&= 1 + \mathcal{O}(\eta^2) + 2\eta(1 - 2p)\alpha. \tag{10}
\end{aligned}
$$

$$
\begin{aligned}
\beta^{(t)} &= \langle \mathbf{v}_{2i}^{(t)}, \mathbf{v}_{2i+1}^{(t)} \rangle \\
&= \frac{1}{r_v^{(t)}} (\beta + \eta \cdot 2(1 - 2p)\alpha) \\
&= \frac{\beta + 2\eta(1 - 2p)\alpha}{1 + 2\eta(1 - 2p)\alpha} \tag{11} \\
&= \beta + \frac{2\eta(1 - 2p)\alpha(1 - \beta)}{1 + 2\eta(1 - 2p)\alpha}. \tag{12}
\end{aligned}
$$

Using the induction assumption that $\alpha^{(t)} > 0$ for all $t \geq 1$, every gradient descent update increases $\beta^{(t)}$ starting at $t = 2$. At equilibrium, $\beta^{(t)}$ can no longer increase, which implies that $\beta^{(\infty)} = 1$.

$$
\begin{aligned}
\alpha^{(t)} &= \left\langle \mathbf{u}_{2i}^{(t)}, \mathbf{v}_{2i}^{(t)} \right\rangle \\
&= \frac{1}{\sqrt{r_u r_v}} \left( \alpha + \eta(1 - 2p)(2 + \beta) + \mathcal{O}(\eta^2) \right) \\
&\approx \frac{\alpha + \eta(1 - 2p)(2 + \beta)}{\sqrt{1 + 4\eta(1 - 2p)\alpha} \sqrt{1 + 2\eta(1 - 2p)\alpha}} \quad \text{(by the given form of } r_v, r_u) \tag{13} \\
&\underset{\text{set } x := \eta(1 - 2p)}{=} \frac{\alpha + x(2 + \beta)}{\sqrt{1 + 4x\alpha} \sqrt{1 + 2x\alpha}} \\
&\approx \frac{\alpha + x(2 + \beta)}{(1 + 2x\alpha)(1 + x\alpha)} \tag{14} \\
&\underset{\eta \text{ small}}{\approx} \frac{\alpha + x(2 + \beta)}{(1 + 3x\alpha)} \\
&= \alpha + \frac{x(2 + \beta - 3\alpha^2)}{(1 + 3x\alpha)} \tag{15}
\end{aligned}
$$

Where Equation 13 follows from the approximation $\sqrt{1 + cx} \approx 1 + \frac{1}{2}cx$ for small $x$ (equivalently, small $\eta$). First, note that by Equation 13 and since $\beta \geq 0$ (including at initialization), we have $\alpha^{(t)} > 0$ for all $t \geq 1$. This also completes the induction step to show that $\beta^{(t)} > 0$ for all $t \geq 2$, using the update equation for $\beta$ (Equation 12). Moreover, as established earlier, at equilibrium we must have $\beta = 1$. Substituting this into Equation 15 further implies that $\alpha = 1$ at equilibrium. Hence, we have successfully proved properties (a) and (b) too.

$\square$

### B.2.2 SUPPORT FOR ASSUMPTION 1 VIA UNIFORM REPRESENTATIONS

Contrastive representations, when normalized, have been proven to converge to a uniform distribution on the unit sphere (Wang & Isola, 2020). Building on this result, we adopt the same assumption to provide additional theoretical support for Assumption 1. The specific form we prove here does not yield exact equality,

$$\mathbb{E}_{p^\pi(s_f|s,a)}[\psi(s_f)] = \phi(s, a),$$

but rather a positive proportionality. But since the multiplicative factor is strictly positive, the argmax in the maximization problem of Theorem 1 remains unchanged and this result is still useful.

**Lemma 2.** *Assuming both high dimensionality ($d \gg 1$) and that, at convergence, the contrastive representations $\psi(s)$ are uniformly distributed on the unit sphere, we have:*

$$\mathbb{E}_{p^\pi(s_f|s,a)}[\psi(s_f)] \approx \frac{1}{d}\exp\left(\frac{1}{2d}\right)\phi(s, a).$$

*Proof.* We begin by noting that, in high dimensions, a uniform distribution on the unit sphere is equivalent to an isotropic Gaussian distribution $\mathcal{N}(0, \frac{1}{d}I_d)$ (see, for instance, Equation 3.15 of Vershynin (2018)):

$$p(\psi) = \frac{1}{(\frac{2\pi}{d})^{d/2}}\exp\left(-\frac{d\|\psi\|^2}{2}\right),$$

where $d$ is the representation dimension.

From the InfoNCE objective (Equation 1), at convergence the representation satisfies

$$\phi(s, a)^\top \psi(s_f) = \log\frac{p^\pi(s_f|s, a)}{p^\pi(s_f)}.$$

Therefore, for the expectation we have

$$\mathbb{E}_{p^\pi(s_f|s,a)}[\psi(s_f)] = \int p^\pi(s_f|s, a)\,\psi(s_f)\,ds_f$$

$$= \int p^\pi(s_f)\,\frac{p^\pi(s_f|s, a)}{p^\pi(s_f)}\,\psi(s_f)\,ds_f$$

$$= \int p^\pi(s_f)\exp\left(\phi(s, a)^\top\psi(s_f)\right)\psi(s_f)\,ds_f.$$

Now, switching variables to $\psi_f := \psi(s_f)$ and substituting the Gaussian density:

$$\mathbb{E}_{p^\pi(s_f|s,a)}[\psi(s_f)] = \int_{\mathbb{R}^d}\frac{1}{(\frac{2\pi}{d})^{d/2}}\exp\left(-\frac{d}{2}\|\psi_f\|^2\right)\exp\left(\phi(s, a)^\top\psi_f\right)\psi_f\,d\psi_f.$$

Completing the square in the exponent:

$$-\frac{d}{2}\|\psi_f\|^2 + \phi(s, a)^\top\psi_f = -\frac{d}{2}\|\psi_f - \frac{\phi(s, a)}{d}\|^2 + \frac{1}{2d}\|\phi(s, a)\|^2.$$

Thus,

$$\mathbb{E}_{p^\pi(s_f|s,a)}[\psi(s_f)] = \exp\left(\frac{1}{2d}\|\phi(s, a)\|^2\right)\int_{\mathbb{R}^d}\frac{1}{(\frac{2\pi}{d})^{d/2}}\exp\left(-\frac{d}{2}\|\psi_f - \frac{\phi(s, a)}{d}\|^2\right)\psi_f\,d\psi_f.$$

The integral above is simply the expectation of a Gaussian random vector with mean $\frac{1}{d}\phi(s,a)$, which equals $\frac{1}{d}\phi(s,a)$. Therefore,

$$\mathbb{E}_{p^\pi(s_f|s,a)}[\psi(s_f)] = \exp\left(\tfrac{1}{2d}\|\phi(s,a)\|^2\right) \frac{1}{d}\phi(s,a).$$

Finally, since in high dimensions the learned $\phi(s,a)$ is unit norm, we obtain

$$\mathbb{E}_{p^\pi(s_f|s,a)}[\psi(s_f)] \approx \frac{1}{d}\exp\left(\tfrac{1}{2d}\right) \phi(s,a).$$

$\square$

### B.3 PROOF OF THEOREM 1

*Proof.* From Assumption 1:
$$\phi(s,a) = \mathbb{E}_{p_\gamma^\pi(s_f|s,a)}[\psi(s_f)].$$

Substituting this into the first term of the SGCRL actor objective (Equation 2), we obtain

$$\max_\pi \; \mathbb{E}_{s\sim p(s),\, a\sim\pi(a|s,g)}\left[\phi(s,a)^\top \psi(g)\right] \;=\; \max_\pi \; \mathbb{E}_{s\sim p(s),\, a\sim\pi(a|s,g)}\left[\mathbb{E}_{p_\gamma^\pi(s_f|s,a)}[\psi(s_f)]^\top \psi(g)\right]$$

Expanding the discounted future state distribution yields

$$= \max_\pi \; \mathbb{E}_{s\sim p(s),\, a\sim\pi(a|s,g)}\left[(1-\gamma)\sum_{s_f}\left(\sum_{t=0}^\infty \gamma^t\, p_t^\pi(s_t=s_f \mid s,a)\right)\psi(s_f)^\top\psi(g)\right]$$

$$= \max_\pi \; \mathbb{E}_{s\sim p(s),\, a\sim\pi(a|s,g)}\left[\sum_{t=0}^\infty \gamma^t\, \mathbb{E}_{p_t^\pi(s_t)}[\psi(s_t)^\top\psi(g) \mid s_0=s, a_0=a]\right]$$

$$= \max_\pi \; \mathbb{E}_{s\sim p(s),\, a\sim\pi(a|s,g)}\left[\mathbb{E}_\pi\left[\sum_{t=0}^\infty \gamma^t\, \psi(s_t)^\top\psi(g) \;\Big|\; s_0=s, a_0=a\right]\right].$$

This is exactly the reinforcement learning reward maximization objective with reward function

$$r(s,a) = \psi(s)^\top\psi(g).$$

Therefore, maximizing the SGCRL objective is equivalent to maximizing the Q-value induced by this reward function. $\square$

### B.4 A STRONGER VERSION OF THEOREM 2 AND PROOF

We now analyze the equilibrium dynamics of the InfoNCE update rule in the following theorem. Claims 1 and 3 of this theorem directly imply the result stated in Theorem 2. For simplicity of notation, we write $\mathbf{u}_i$ in place of $\phi(s_i, a_i)$ and $\mathbf{v}_i$ in place of $\psi(sf_i)$.

**Theorem 3** (InfoNCE representations at equilibrium)**.** *Let $\mathbf{z}\in\mathbb{R}^d$ be a fixed unit vector, with $d\gg 1$. Let $\{\mathbf{u}_i\}_{i=1}^n$ and $\{\mathbf{v}_i\}_{i=1}^n \subset \mathbb{R}^d$ be anchor and future embeddings, initialized as:*

$$\mathbf{u}_i^0 = c\,\mathbf{z} + \zeta_i^0, \quad \mathbf{v}_i^0 = c\,\mathbf{z} + \kappa_i^0$$

*where $\zeta_i^0, \kappa_i^0 \overset{i.i.d.}{\sim} \mathcal{N}\left(0, \frac{1-c^2}{d}I_d\right)$ and $c$ is a scalar. Suppose these vectors are updated via gradient descent on the backward (or forward) InfoNCE loss with batch size $N\gg 1$ and step size $\eta>0$, followed by unit-norm normalization. We assume $\eta$ is sufficiently small.*

*Then, with high probability over the initialization, the dynamics satisfy the following:*

1. *At every step t, each representation decomposes as*

$$\mathbf{u}_i^t = c^t \mathbf{z} + \zeta_i^t, \quad \mathbf{v}_i^t = c^t \mathbf{z} + \kappa_i^t,$$

   *where $\zeta_i^t, \kappa_i^t \perp \mathbf{z}$ and $c^t$ is the same for all $i$.*

2. *At fixed point (i.e., when all the gradients are zero), $\mathbf{u}_i^{(\infty)} = \mathbf{v}_i^{(\infty)}$, $\forall i$ , and $\langle \mathbf{u}_i^{(\infty)}, \mathbf{v}_j^{(\infty)} \rangle = 0, i \neq j$*

3. *At fixed point: $c^{(\infty)} = 0$, i.e., all representations become orthogonal to $\mathbf{z}$ as $t \to \infty$.*

*Proof.* We establish the above results, together with three additional claims regarding the InfoNCE update dynamics at equilibrium, via induction. At iteration $t$, we decompose each representation as

$$\mathbf{u}_i^t = c^{(t)} \mathbf{z} + \zeta_i^t, \qquad \mathbf{v}_i^t = c^{(t)} \mathbf{z} + \kappa_i^t,$$

where the residuals $\zeta_i^t$ and $\kappa_i^t$ are orthogonal to the unit vector $\mathbf{z}$. We define the following quantities and note that they are invariant with respect to the choice of index $i$ (or $i \neq j$ where applicable):

(a) $\langle \mathbf{u}_i^t, \mathbf{z} \rangle = \langle \mathbf{v}_i^t, \mathbf{z} \rangle = c^{(t)}$ for all $i$.

(b) $\alpha^{(t)} := \langle \zeta_i^t, \kappa_i^t \rangle$.

(c) $\lambda^{(t)} := \langle \zeta_i^t, \kappa_j^t \rangle = \langle \zeta_i^t, \zeta_j^t \rangle = \langle \kappa_i^t, \kappa_j^t \rangle = 0$ for all $i \neq j$ and for all $t$.

(d) $r^{(t)} := \|\zeta_i^t\|^2 = \|\kappa_i^t\|^2$ for all $i$.

**Base case ($t = 0$):** By initialization, $\zeta_i^0, \kappa_i^0 \sim \mathcal{N}(0, \frac{1-c^2}{d} I_d)$ i.i.d., and $\zeta_i^0, \kappa_i^0 \perp \mathbf{z}$. In high dimensions, with probability 1 these vectors are all orthogonal to each other therefore $\lambda^{(0)} = \alpha^{(0)} = 0$ and $c^0$ is the same for all vectors by construction. And $\|\zeta_i^0\| = \|\kappa_i^0\| = 1 - c^2$ by construction. Hence, all four properties a,b,c,d hold at $t = 0$.

**Inductive step:** Assume the properties hold at time $t - 1$. We now prove they also hold at time $t$.

We first prove it for the case that representations are normalized and for the backward InfoNCE loss, the proof for the forward InfoNCE loss is exactly the same due to the symmetry at initialization, which, as we will see later, is maintained through all the updates. The backward InfoNCE updates with normalization are given by:

$$\hat{\mathbf{u}}_i^t = \mathbf{u}_i^{t-1} + \eta \left( \mathbf{v}_i^{t-1} - \sum_j p_{ij} \mathbf{v}_j^{t-1} \right) \tag{16}$$

$$\mathbf{u}_i^t = \frac{\hat{\mathbf{u}}_i^t}{\|\hat{\mathbf{u}}_i^t\|}, \qquad \text{similarly for } \mathbf{v}_i^t \tag{17}$$

where $p_{ij}^{(t-1)} = \frac{\exp(\langle \mathbf{u}_i^{t-1}, \mathbf{v}_j^{t-1} \rangle)}{\sum_k \exp(\langle \mathbf{u}_k^{t-1}, \mathbf{v}_j^{t-1} \rangle)}$.

Due to symmetry at time $t - 1$, we have:

$$p_{ij}^{(t-1)} = p_{ji}^{(t-1)}, \quad p_{ii}^{(t-1)} =: p^{(t-1)} \quad \text{for all } i, j$$

so the matrix $P^{(t-1)}$ is symmetric, with equal diagonals and exchangeable off-diagonals. For ease of notation we use $p := p_{ii}^{(t-1)}$ and $q := p_{ij}^{(t-1)}$. Note that $p + (N - 1)q = 1$

**Property (a), (d): projection onto z and norm** From the updates and the fact that all $c^{(t-1)}$ are equal, we get:

$$\left\langle \mathbf{v}_i^{t-1} - \sum_j p_{ij} \mathbf{v}_j^{t-1}, \mathbf{z} \right\rangle = c^{(t-1)} - \sum_j p_{ij} c^{(t-1)} = 0$$

Thus,

$$\langle \hat{\mathbf{u}}_i^t, \mathbf{z} \rangle = \langle \mathbf{u}_i^{t-1}, \mathbf{z} \rangle = c^{(t-1)}, \quad \Rightarrow \quad \langle \mathbf{u}_i^t, \mathbf{z} \rangle = \frac{c^{(t-1)}}{\|\hat{\mathbf{u}}_i^t\|} =: c^{(t)} \tag{18}$$

and the same holds for $\mathbf{v}_i^t$. In order to prove (a), we need to show that $\|\mathbf{u}_i\|$ and $\|\mathbf{v}_i\|$ are equal and invariant to the index for any $t$. (For ease of notation we use $\alpha, \lambda, r$ instead of $\alpha^{(t-1)}, \lambda^{(t-1)}, r^{(t-1)}$.)

$$\begin{aligned}
\|\hat{\mathbf{u}}_i^t\|^2 &= \left\| c^{(t-1)}\mathbf{z} + \zeta_i^{t-1} + \eta\left( (1-p)\kappa_i^{t-1} - \sum_{j \neq i} q\kappa_j^{t-1} \right) \right\|^2 \\
&= \left( c^{(t-1)} \right)^2 + r + 2\eta(1-p)\alpha - 2\eta(N-1)q\,\lambda \\
&= 1 + 2\eta(1-p) \cdot \alpha + \mathcal{O}(\eta^2)
\end{aligned} \tag{19}$$

We used the fact that due to normalization $(c^{(t-1)})^2 + r = 1$, we also used $\lambda = 0$.

It is straightforward to verify that the expression for $\|\hat{\mathbf{u}}_i^t\|^2$ is independent of the choice of index $i$. Furthermore, if we expand $\|\hat{\mathbf{v}}_i^t\|^2$, we encounter the same number of matched pairwise or cross-term inner products, with only the order of terms being swapped. As a result, we obtain the same expression.

We therefore denote this common quantity by

$$L^{(t-1)} := \|\hat{\mathbf{u}}_i^t\|^2 = \|\hat{\mathbf{v}}_i^t\|^2 .$$

Therefor, it follows from Equation 18 that $\langle \mathbf{u}_i^t, \mathbf{z} \rangle$ is identical for all representations. Since all these representations share the same norm and identical projection onto $\mathbf{z}$ (i.e., the parallel component), it must be that their orthogonal components—namely, $\zeta_i^t$ and $\kappa_i^t$—also have equal norms. Hence, condition (d) is satisfied too, this also ends the proof for claim 1 of the theorem statement.

**Properties (b), (c): Matching and cross inner product** We expand:

$$\zeta_i^t = \frac{1}{L^{(t-1)}} \left( \zeta_i^{t-1} + \eta\left( (1-p)\kappa_i^{t-1} - \sum_{j \neq i} q\kappa_j^{t-1} \right) \right)$$

$$\kappa_i^t = \frac{1}{L^{(t-1)}} \left( \kappa_i^{t-1} + \eta\left( (1-p)\zeta_i^{t-1} - \sum_{j \neq i} q\zeta_j^{t-1} \right) \right)$$

Then:

$$\begin{aligned}
\alpha^{(t)} = \langle \zeta_i^t, \kappa_i^t \rangle &= \frac{1}{(L^{(t-1)})^2} \left[ \alpha + 2\eta(1-p)r - 2\eta(N-1)\lambda q + \mathcal{O}(\eta^2) \right] \\
&= \frac{1}{(L^{(t-1)})^2} [\alpha + 2\eta r(1-p) + \mathcal{O}(\eta^2)]
\end{aligned}$$

This expression is independent of the choice of index $i$; this proves statement (b). Similarly, we can evaluate $\langle \zeta_i^{(t)}, \kappa_j^{(t)} \rangle$ and $\langle \zeta_i^{(t)}, \zeta_j^{(t)} \rangle$ for $i \neq j$:

$$\zeta_i^t = \frac{1}{L^{(t-1)}} \left( \zeta_i^{t-1} + \eta\left( (1-p)\kappa_i^{t-1} - \sum_{l \neq i} q\kappa_l^{t-1} \right) \right)$$

$$\zeta_j^t = \frac{1}{L^{(t-1)}} \left( \zeta_j^{t-1} + \eta\left( (1-p)\kappa_j^{t-1} - \sum_{l \neq j} q\kappa_l^{t-1} \right) \right)$$

$$\kappa_j^t = \frac{1}{L^{(t-1)}} \left( \kappa_j^{t-1} + \eta\left( (1-p)\zeta_j^{t-1} - \sum_{l \neq j} q\zeta_l^{t-1} \right) \right)$$

$$\langle \zeta_i^t, \zeta_j^t \rangle = \frac{1}{(L^{(t-1)})^2}[\lambda + 2\eta(1-p)\lambda - 2\eta q(N-2)\lambda - 2\eta q\alpha + \mathcal{O}(\eta^2)]$$

$$\underset{N \text{ Large}}{=} \frac{1}{(L^{(t-1)})^2}[\lambda + 2\eta(1-p)\lambda - 2\eta q(N-1)\lambda + \mathcal{O}(\eta^2)]$$

$$\underset{p+(N-1)q=1}{=} \frac{1}{(L^{(t-1)})^2}[\lambda + \mathcal{O}(\eta^2)]$$

$$\langle \zeta_i^t, \kappa_j^t \rangle = \frac{1}{(L^{(t-1)})^2}[\lambda + 2\eta(1-p)\lambda - 2\eta q(N-2)\lambda - 2\eta qr + \mathcal{O}(\eta^2)]$$

$$\underset{N \text{ Large}}{=} \frac{1}{(L^{(t-1)})^2}[\lambda + 2\eta(1-p)\lambda - 2\eta q(N-1)\lambda + \mathcal{O}(\eta^2)]$$

$$\underset{p+(N-1)q=1}{=} \frac{1}{(L^{(t-1)})^2}[\lambda + \mathcal{O}(\eta^2)]$$

Since $N$ is large and $p, q \geq 0$ with $p + (N-1)q = 1$, it follows that $q \approx 0$. We observe that these inner products are also independent of the specific choice of $i$ and $j$, due to the same underlying symmetry. Moreover $\langle \zeta_i^{(t)}, \kappa_j^{(t)} \rangle$ and $\langle \zeta_i^{(t)}, \zeta_j^{(t)} \rangle$ are all zero given that $\lambda$ is zero (induction) hence (c) is also proved.

This completes the inductive step, i.e., the proof for a, b, c, d.

Now we assess how $\alpha^{(t)}$ and $r^{(t)}$ change to prove claims 2, 3 of the theorem statement.

$$\alpha^{(t)} = \frac{1}{(L^{(t-1)})^2}[\alpha + 2\eta r(1-p) \cdot r + \mathcal{O}(\eta^2)] \tag{20}$$

$$\approx \frac{\alpha + 2\eta(1-p) \cdot r}{1 + 2\eta(1-p) \cdot \alpha}$$

$$= \alpha + \frac{2\eta(1-p)r \cdot \left(1 - \alpha\frac{\alpha}{r}\right)}{1 + 2\eta(1-p) \cdot \alpha}$$

$$\underset{\alpha \leq 1}{\geq} \alpha + \frac{2\eta(1-p)r \cdot \left(1 - \frac{\alpha}{r}\right)}{1 + 2\eta(1-p) \cdot \alpha} \tag{21}$$

$$\geq \alpha + \frac{2\eta(1-p)r \cdot (1 - \frac{\alpha}{r})}{1 + 2\eta(1-p) \cdot \alpha} \tag{22}$$

$$\geq \alpha$$

Since the denominator is positive and $1 - \alpha/r \geq 0$. $\alpha$ only stops growing if the equality holds i.e., $\alpha/r = 1$,

$$\frac{\langle \zeta_i^{(t-1)}, \kappa_i^{(t-1)} \rangle}{\|\zeta_i^{(t-1)}\| \, \|\kappa_i^{(t-1)}\|} = 1 \quad \Longleftrightarrow \quad \cos\left(\zeta_i^{(t-1)}, \kappa_i^{(t-1)}\right) = 1,$$

so as long as $\cos\left(\zeta_i^{(t-1)}, \kappa_i^{(t-1)}\right) < 1$, $\alpha^{(t)}$ increases, at equilibrium its growth should be stopped and that means the alignment of each positive pair increases strictly, and positive pairs keep aligning until they are fully aligned. This result, along with the fact that $\lambda^{(t)} = 0$, $\forall t$, which we proved before, completes the proof for claim 2 of the theorem statement.

Finally, we prove Claim 3 of the theorem by showing that, at equilibrium, $r = 1$, i.e., the squared norm of the component orthogonal to $\mathbf{z}$. Since each representation is normalized, this immediately implies that the parallel component must vanish.

$$r^{(t)} = r + 2\eta(1-p)\alpha$$

Since $\alpha^{(0)} = 0$ and, by Equation 21, $\alpha^{(t)}$ increases strictly (Note that due to norm 1 constraint on the representations $p$ can never approach 1) until full alignment, it follows that $\alpha^{(t)} > 0$ for all $t \geq 1$. Consequently,

$$r^{(t)} > r^{(t-1)} \qquad \text{for all } t \geq 2.$$

Thus, the sequence $\{r^{(t)}\}$ is strictly increasing until it saturates at the unit-norm bound. □

### B.5 IMPLICATION OF THEOREM 2 IN FUNCTION APPROXIMATION SETTING

Theorem 2 shows that an agent trained with contrastive RL can rule out regions previously visited where the goal was not found, as indicated by low $\psi$-similarity. At first glance, this mechanism might appear inefficient in continuous settings, where the state space is infinite. However, in the function approximation regime, we expect the critic network to generalize in two important ways: (1) states that are far from the goal in the underlying state space should also be represented as far from the goal in the embedding space, making them unattractive to explore; and (2) if a region is ruled out as not containing the goal (low $\psi$-similarity), nearby states should likewise be ruled out. Thus, with function approximation, this mechanism extends naturally to infinitely many states. Indeed, as we have observed in previous work (Liu et al., 2025), SGCRL is capable of solving long-horizon planning tasks in continuous environments. Also, we refer the reader to Appendix D.6 for two experiments that show SGCRL in continuous setting explores the environment efficiently and avoids non goal-unrelated states.

## C EXPERIMENTAL DETAILS

Table 1: SGCRL Hyperparameters.

| hyperparameter | value |
| --- | --- |
| **Standard SGCRL** (Liu et al., 2025) | |
| batch size | 256 |
| learning rate | 3e-4 |
| discount | 0.99 |
| actor target entropy | 0 |
| hidden layers sizes (policy, critic) | (256, 256) |
| initial random data collection | 10,000 transitions |
| replay buffer size | 1e6 |
| samples per insert[1] | 256 |
| representation dimension ($\dim(\phi(s,a)), \dim(\psi(s_g))$) | 64 |
| actor minimum std dev | 1e-6 |
| **Tabular SGCRL** | |
| batch size | 128 |
| learning rate | 1e-2 |
| discount | 0.99 |
| initial random data collection | False |
| replay buffer size | 1e3 |
| representation dimension ($\dim(\psi(s_g))$) | 16 |

[1] How many times is each transition used for training before being discarded.

## D ADDITIONAL EXPERIMENTS

### D.1 TABULAR SINGLE GOAL EXPLORATION FALLS WITHIN A CLASS OF CLASSICAL EXPLORATION ALGORITHMS

Through comparison with tabular exploration methods, we investigate whether SGCRL employs classical exploration strategies or explores in a unique way. Historically, statistically-efficient exploration algorithms broadly employ one of two strategies: optimism in the face of uncertainty (Kearns & Singh, 2002; Brafman & Tennenholtz, 2002; Kakade, 2003; Strehl et al., 2009; Jaksch et al., 2010; Jin

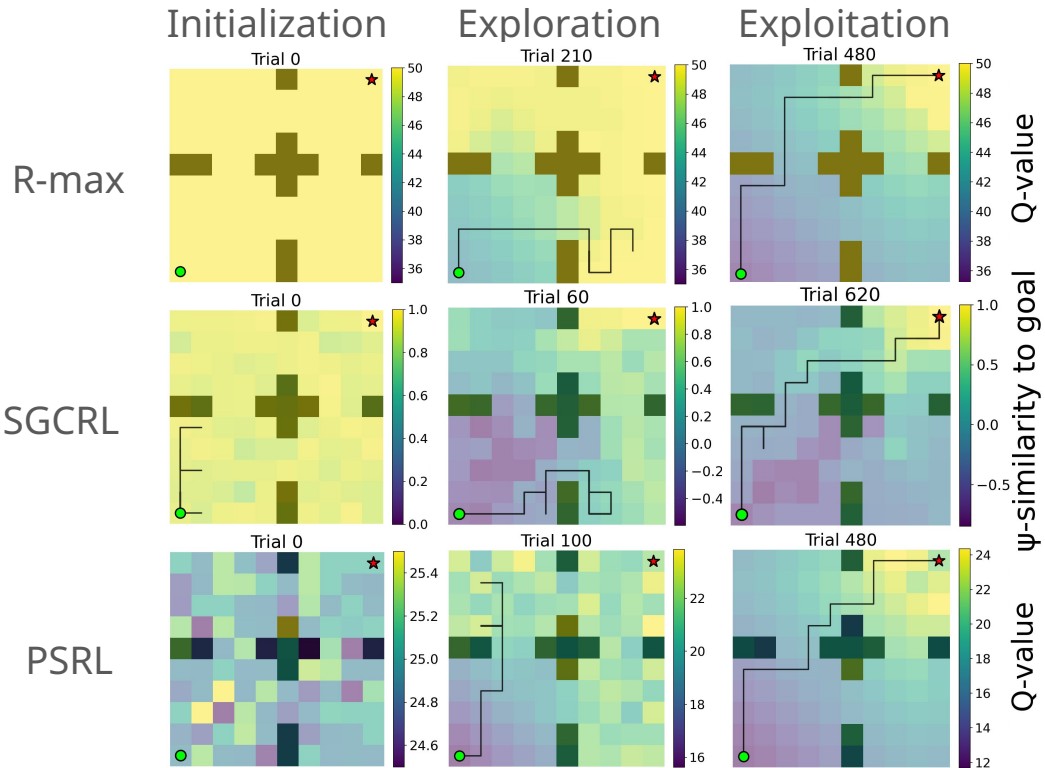

Figure 6: SGCRL shares characteristics with R-max and PSRL

et al., 2018) or posterior sampling (Osband et al., 2013; Osband & Van Roy, 2014; Abbasi-Yadkori & Szepesvari, 2014; Agrawal & Jia, 2017; Osband & Van Roy, 2017). One representative delegate of each is R-MAX (Brafman & Tennenholtz, 2002) and Posterior Sampling for Reinforcement Learning (PSRL) (Strens, 2000; Osband et al., 2013), respectively. Curiously, instances in either camp may admit a unified regret analysis through the construction of confidence sets (Russo & Van Roy, 2013; Osband & Van Roy, 2014; Lu & Van Roy, 2019), collections that hold the true ground-truth hypothesis with high probability and (by virtue of a good exploration strategy) can be shown to have shrinking widths as data accumulates.

Empirically, we observe that SGCRL operates according to this same unified perspective, maintaining an implicit collection of hypothesized goal states with optimistically-inflated values and progressively refining this confidence set through targeted exploration. The algorithm's goal-conditioned representation learning creates an initial landscape where many states appear promising (similar to the goal). By visiting these candidate states, the algorithm systematically winnows this set until it finds the true goal state (see Fig. 6). Visualizing training checkpoints for all three algorithms, we find that, due to optimistic initialization, SGCRL progressively explores candidate states like R-MAX until it finds the goal, at which point it demonstrates the rapid exploitation of PSRL (upon identifying the true underlying environment).

## D.2 SGCRL ACHIEVES MODEST SUCCESS WHEN TRAINED WITH DATA COLLECTED BY ANOTHER ALGORITHM

In this experiment, we investigate the role of the actor data collection algorithm in SGCRL performance. We conduct yoked experiments in which we perform the SGCRL representation updates using data collected by another algorithm (e.g. PSRL, R-MAX) running independently in parallel. We also conduct a yoked control experiment in which SGCRL learns using the data collected by another independently initialized SGCRL agent running in parallel. We find that when we yoke SGCRL to PSRL or R-MAX, it learns to reach the goal, though not consistently (Fig. 7a, 7b). When we yoke SGCRL to another SGCRL instance, both agents learn to consistently solve the task (Fig. 7c). These results imply that SGCRL's can learn somewhat useful representations when trained on data

collected by another algorithm, but still attains the best performance with single-goal data collection. Contrary to previous work in both cognitive science (Markant & Gureckis, 2014; 2010) and deep RL (Ostrovski et al., 2021), the success of the SGCRL-SGCRL yoked experiment implies that online interaction with the environment is not necessary.

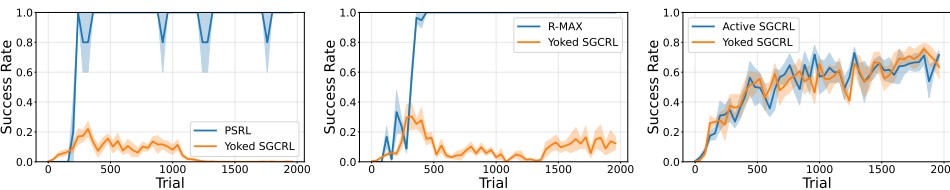

(a) SGCRL yoked with PSRL  (b) SGCRL yoked with R-MAX  (c) SGCRL yoked with SGCRL

Figure 7: (a,b) SGCRL acheives modest success when trained on data collected by PSRL or R-MAX. (c) SGCRL succeeds consistently when trained on data collected by a different SGCRL initialization. Evaluation curves show the performance of the yoked SGCRL agent policy trained on data collected by another agent.

### D.3    SGCRL FAILS TO EXPLORE WITHOUT REPRESENTATIONS

In this ablation experiment, we investigate whether representations are important for strategic exploration. We ablate the representations by implementing a version of tabular SGCRL without representations. Rather than parameterizing $\psi$ with a 16-dimensional vector for each state, we simply maintain an $|\mathcal{S}| \times |\mathcal{S}|$ table of scalar values for the $\psi$-similarity of each state, goal pair. We learn the values in this table using the same InfoNCE updates and the rest of the method is exactly the same as tabular SGCRL. We find that by ablating vectorized representations (Fig. 8), the agent fails to explore effectively, requiring on the order of 100x more samples to find the goal and converging to a low success rate. The results of this experiment imply that SGCRL representations, whether approximated by a neural network or a vector, are important for strategic and efficient exploration.

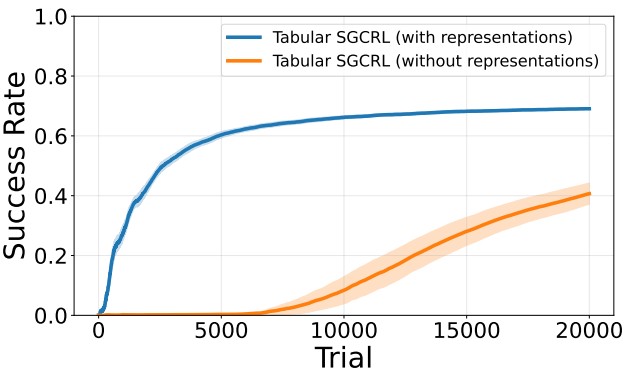

Figure 8: Replacing the vectorized representations in tabular SGCRL with a lookup table results in slower exploration.

### D.4    SINGLE-GOAL EXPLORATION IN ROBOTIC MANIPULATION TASKS.

We find that the characterization of SGCRL detailed in Section 4 holds in a Sawyer robotic manipulation task where the agent must pick up a block and place it in a bin (Yu et al., 2020). During early stages of training, the agent moves the robotic end-effector towards regions of high representational similarity to the goal. Subsequently, the representational goal similarity of these frequently visited regions decreases, and the agent visits new regions (see Fig. 9). These observational results suggest that our characterization of single-goal exploration generalizes beyond 2D navigation tasks.

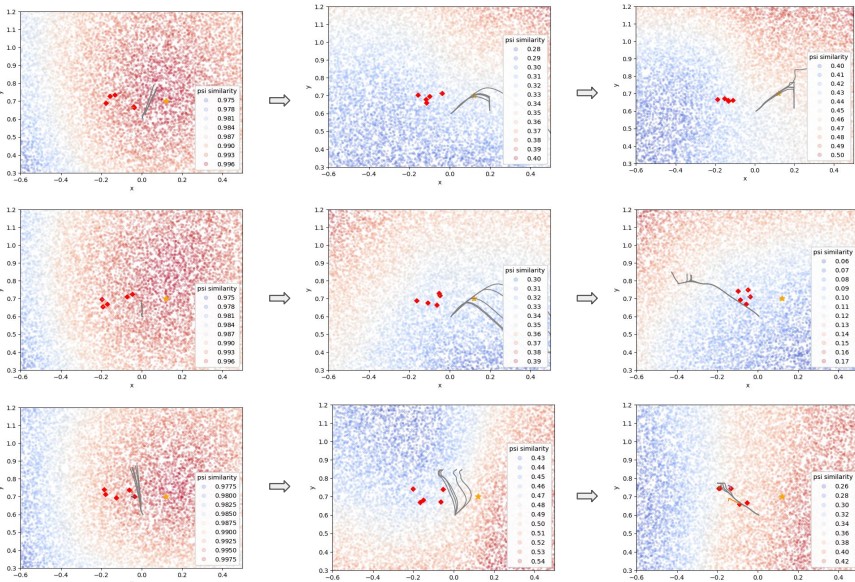

Figure 9: XY cross section of representational goal similarity for the Sawyer Bin environment. Each row represents checkpoints throughout training for different training seeds. The gray lines show the trajectory of the end-effector across 5 episodes. The agent moves the end-effector towards regions of high $\psi$-similarity to the goal, and then those regions subsequently develop low $\psi$-similarity, driving continued exploration to new areas.

## D.5   SINGLE-GOAL DATA COLLECTION IS ESSENTIAL FOR EXPLORATION-ENCOURAGING REPRESENTATIONS

In this section, we analyze the role of single-goal exploration in forming representations that promote exploration even before the goal is discovered. We also examine the behavior of a single-goal exploration agent when the goal is unreachable. Interestingly, the agent still engages in broad exploration, eventually covering the entire state space, and at convergence, it settles into a random walk across the maze since the goal is never found. To conduct experiments, we use the simplified tabular model of SGCRL without neural networks introduced in Section 5.

As we saw earlier, the central mechanism that enables SGCRL's effectiveness is precisely its ability to drive the $\psi$-similarity of non-goal regions toward zero. This ensures that previously visited goal-free states are ruled out, forcing the agent to focus on unexplored areas. Theorem 2 establishes this formally, showing that contrastive representations in SGCRL implement exactly this property.

In practice, however, the assumptions of Theorem 2 – for instance, symmetric initialization – do not necessarily hold. For instance, in the four-room environment, the agent begins in the bottom-left room and gradually visits new ones. Adding states from newly visited rooms to the replay buffer alters the updates of earlier rooms, breaking the symmetry assumption and eliminating the guarantee of orthogonality between room representations. In this section, we address two key questions: 1) Does the phenomenon of decreasing $\psi$-similarity persist in more realistic settings? 2) If so, what are the underlying dynamics, and are they unique to SGCRL's goal-directed data collection, or can other exploration strategies achieve the same effect?

Evolving the representations, as shown in Figure 4, demonstrates that even in realistic scenarios the representations of visited rooms drift farther from the goal representation at $(0, 0, 1)$ on the $z$-axis. This behavior is intuitive: first, within a single room, the shared component aligned with $\psi(g) = \mathbf{z}$ is essentially wasted energy for contrastive learning. To achieve a stronger contrastive loss among states in the same room, the $\mathbf{z}$-component of their representations is dampened (refer to Theorem 2). Second, as the agent explores more and begins visiting new rooms, the introduction of these new room representations pushes the older room representations even further away from the goal. This

happens because newly visited rooms are initialized close to $\mathbf{z}$, so in order to maintain contrast, older rooms are better off shifting downward and away from it.

This observation naturally leads to the following question:

- **RQ1.** What happens if the representations of some areas in the new rooms are not initialized close to $\mathbf{z}$? Do we still see the decreasing $\psi$-similarity trend of visited states which is essential for exploration?

- **RQ2.** Is single-goal exploration data collection necessary for $\psi$-similarity reduction in visited states? What does the representation evolution look like if we instead use a multi-goal exploration data collection policy that collect data by sampling a new exploratory goal in the beginning of every episode?

To explore this, we designed a new experiment. In this setting, the representations of a small patch in the top-left and bottom-right rooms is initialized orthogonal to $\mathbf{z}$ (in order to address RQ1), while the rest of the state representations are initialized as $\mathbf{z}$ plus small random noise (see Figure 10). The agent starts in the bottom-left room (marked by the green dot).

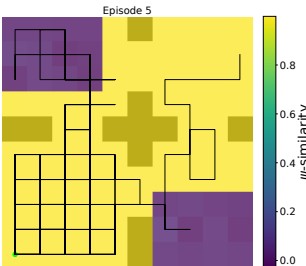

Figure 10: SGCRL data collection vs random goal data collection. At initialization, all state representations are a noisy version of the imaginary goal representation ($\mathbf{z}$) while the two small patches in the top left room and the bottom right room are initialized with an initialization orthogonal to $\mathbf{z}$.

Apart from representation initialization, the experimental setup is the same as the imaginary goal experiment of Section 5.1. i.e., $\psi(g) = \mathbf{z}$ is a random Gaussian vector that does not correspond to the representation of any actual maze state, simulating a scenario where many representation updates occur without the agent ever observing the true goal. This experimental setup also allows us to address another interesting side question:

- **RQ3.** How does the agent behave during early training and after convergence when the goal is not feasible in the environment?

We start by answering RQ1 and RQ2 through comparing the following two data-collection strategies:

1. **Single-goal exploration** – actions are selected according to the greedy policy:

$$a_t \sim \frac{1}{Z(s_t)} \exp\left(\frac{1}{\tau}\psi(p(s_t, a_t))^\top \mathbf{z}\right).$$

2. **Random-goal exploration** – actions are chosen based on:

$$a_t \sim \frac{1}{Z(s_t)} \exp\left(\frac{1}{\tau}\psi(p(s_t, a_t))^\top \mathbf{y}_e\right), \mathbf{y}_e \sim N(0, I_d)$$

Where $p(s_t, a_t)$ is the environment dynamic that outputs $s_{t+1}$; moreover $\mathbf{y}_e$ is a Gaussian-sampled goal embedding drawn anew at each episode. This corresponds to the widely used convention in earlier works Andrychowicz et al. (2017); Eysenbach et al. (2022) where exploration is guided by sampling from a distribution of potential goals to learn more effectively about the environment.

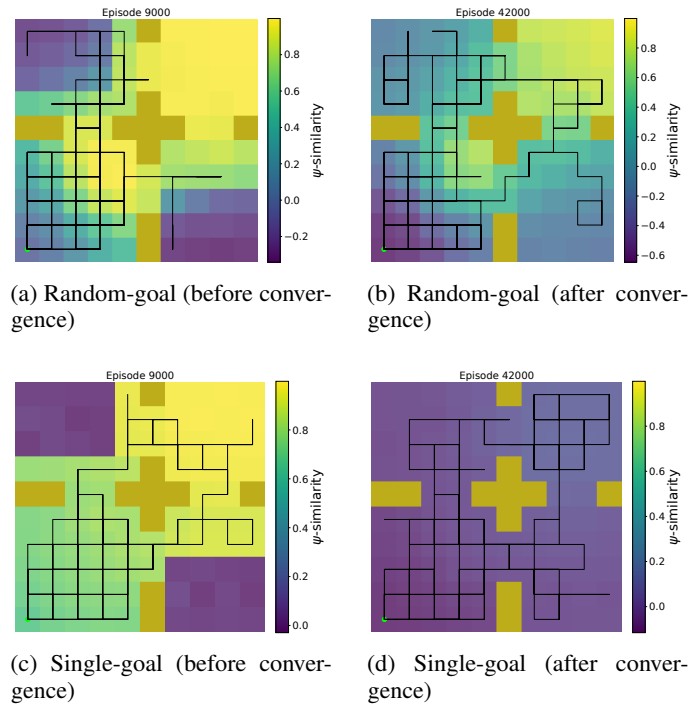

(a) Random-goal (before convergence)

(b) Random-goal (after convergence)

(c) Single-goal (before convergence)

(d) Single-goal (after convergence)

Figure 11: Comparison of representation evolution under two data collection strategies: (Top row) random goal exploration and (Bottom row) SGCRL. Solid lines demonstrate the agent trajectory across 5 episodes.

Both data collection strategies start from the same initial representations. We then compare how the representations evolve under these two different data collection strategies.

As demonstrated in Figure 11, at episode 9000—before the representations have fully converged—the single-goal strategy naturally avoids the small dark patches. Because these patches have lower $\psi$-similarity compared to their surroundings, the policy does not visit them. Instead, it focuses on areas with higher $\psi$-similarity. As a result, the representations of the most frequently updated states consistently evolve by moving farther away from the goal, allowing them to form stronger contrasts with the newly visited states that are closer to the goal in representation.

In contrast, the random goal exploration strategy does not avoid the dark patches (Figure 12a). By visiting these areas, it adds their states to the replay buffer, which in turn forces the older room representations to contrast with these new states. This dynamic prevents the older representations from drifting away from the goal. Indeed, even after convergence, the $\psi$-similarity under random goal exploration fail to approach zero, as shown in Figure 12b.

Figure 12 further visualizes this evolution by projecting the representations into three dimensions using PCA. The key takeaway is that single goal greedy data collection provides an implicit structural benefit: by consistently moving toward regions of higher $\psi$-similarity, it pushes the representations of previously visited areas farther from the goal. In doing so, it shapes the representations of visited, non-goal regions in a way that naturally encourages further exploration.

**Agent behavior when the goal is unreachable** Finally, we address RQ3 by examining the agent's behavior during both early and late training. The single-goal exploration agent is consistently drawn toward new states, but never toward the dark patch, since the representation of that region lies far from $\psi(g)$. As training progresses without ever reaching the goal, the representations of all states gradually become orthogonal to the goal, yielding a uniform $\psi$-similarity of zero across the maze. In this regime, the agent performs an unstructured random walk indefinitely (Figure 12d). By contrast, the multi-goal exploration agent exhibits more diverse behavior, since changing goals occasionally directs it toward states that are initially far from $\psi(g)$. However, because it fails to uniformly reduce

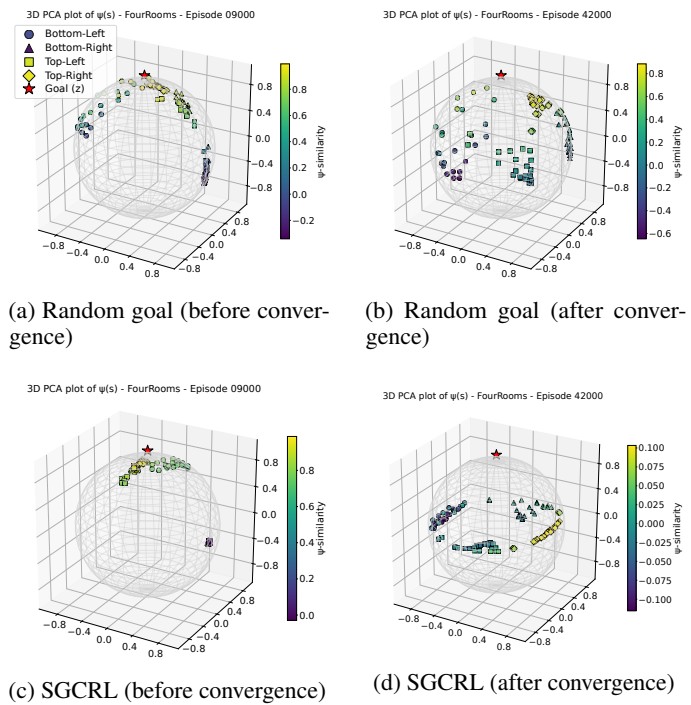

(a) Random goal (before convergence)

(b) Random goal (after convergence)

(c) SGCRL (before convergence)

(d) SGCRL (after convergence)

Figure 12: Comparison of representation evolution under two data collection strategies, with representations projected into 3D using PCA. (Top row) random goal exploration and (Bottom row) SGCRL.

$\psi$-similarity across the entire state space, it often becomes trapped in small regions that retain slightly higher similarity values (Figure 12b).

## D.6   SINGLE-GOAL EXPLORATION EQUIPPED WITH NEURAL NETWORKS AVOIDS EXHAUSTIVE STATE SPACE SEARCH

Our earlier analysis in Sections 4 and 5 showed that single-goal exploration operates as an optimization process: it begins by treating every state as a potential goal and, through contrastive learning, progressively prunes away non-goal states. In Appendix D.1, we demonstrate the similarity of SGCRL to optimism-based and uncertainty-based methods such as R-MAX and PSRL.

However, in the worst case, these classical tabular exploration methods require an exhaustive search over the entire state space $\mathcal{O}(\mathcal{S})$, which is infeasible in continuous settings. While this limitation holds for our simplified tabular SGCRL model without neural networks, we conjecture that the standard SGCRL algorithm with neural network approximation ($\psi(s) = F_\theta(s)$) is less susceptible to exhaustive search. Due to generalization, it is plausible that the reduction in $\psi$-similarity for visited states also extends to nearby states, thereby pruning not only the states actually visited but potentially additional states that would otherwise lead to unproductive exploration.

To test our conjecture, we designed two experiments that evaluate state coverage in the presence of distracting or irrelevant dimensions. The first setup introduces exogenous noise (noisy TV), while the second introduces a controllable but irrelevant dimension. Together, these experiments probe whether SGCRL avoids exhaustive search over parts of the state space that do not contribute to solving the task. Our results suggest that SGCRL focuses its exploration on task-relevant regions of the state space rather than expanding coverage indiscriminately. In contrast, the novelty-driven method PPO+RND, which incorporates an optimism-based novelty bonus, often expends effort exploring irrelevant states. This contrast highlights the practicality of SGCRL in continuous domains, where exhaustive search is infeasible and where other optimism-based approaches may struggle.

Table 2: State coverage (grid) comparison between PPO+RND and SGCRL. Lower coverage in the noisy-TV and irrelevant-dimension experiments indicates robustness to distraction and irrelevant state dimensions.

| Method | Noisy TV ↓ | Irrelevant Dimension ↓ | Four-Room (no noise) |
|---|---|---|---|
| PPO+RND | 0.91 | 0.65 | 0.82 |
| SGCRL | **0.36** | **0.40** | 0.90 |

**Baseline.** As a baseline, we consider a popular optimism-driven exploration method in continuous control: PPO combined with Random Network Distillation (PPO+RND) (Burda et al., 2019; Schulman et al., 2017). In addition to the goal-conditioned extrinsic reward $\mathbb{1}(s = g)$, this method incorporates an intrinsic reward based on the mean squared error between the predictions of two networks, $A$ and $B$, where $B$ is slowly distilled into $A$. This prediction error serves as a measure of novelty, assigning higher rewards to states that have not been frequently visited. The agent is trained using PPO to maximize a weighted sum of the extrinsic reward (scale 2) and intrinsic reward (scale 1). We run this baseline for 300k environment steps. All results are averaged over at least 5 seeds of randomness.

**Environment.** We consider a continuous $11 \times 11$ four-room maze. The agent starts in the top-left corner, and the goal is located in the bottom-right corner.

### D.6.1 NOISY TV EXPERIMENT

**Setup.** Following prior work on exploration, we evaluate whether algorithms fall into the so-called *noisy TV trap*, a scenario where an agent may be distracted by uncontrolled stochastic signals rather than making meaningful progress toward the goal (Burda et al., 2019; Pathak et al., 2017b). To simulate this phenomenon, we augment the bottom-left room of the four-room maze with an additional stochastic dimension $z$, sampled uniformly from $[0, 11]$ at every step. In all other rooms, the $z$-value is fixed at zero. This setup creates an exogenous source of noise that the agent cannot influence. The purpose of this experiment is to compare the state coverage achieved by PPO+RND and SGCRL in the presence of such uncontrollable noise.

**Results.** We compare SGCRL and PPO+RND on task success and exploration behavior during training. In the base environment, SGCRL achieves a success rate of 98%, while PPO+RND reaches 90%. In the noisy-TV setting, success rate decreases by about 8% for SGCRL and 11% for PPO+RND, indicating that SGCRL is slightly more robust to the noisty-TV problem. However, task success alone does not reveal exploration efficiency. We additionally measure state coverage of both algorithms by discretizing the continuous maze into grid cells and computing the fraction of visited cells. Table 2 reports results after 300k environment steps.

In the standard setting, which corresponds to a grid of 121 states, both methods achieve broad coverage (0.90 for SGCRL vs. 0.82 for PPO+RND). In the noisy-TV setting, where the grid expands to 371 states due to the added noisy dimension, a clear divergence emerges: PPO + RND attains much higher coverage (0.91) compared to SGCRL (0.36). This difference reflects the tendency of PPO + RND to overexplore irrelevant states. The noisy TV introduces additional dimensions with high intrinsic novelty, drawing PPO+RND to explore uninformative states. In contrast, SGCRL maintains relatively low coverage (0.36), indicating that the additional noisy-TV states do not cause it to overexplore uninformative states.

Moreover, we plot the average fraction of each episode spent in the bottom-left room (the noisy-TV room) in Figure 27, where episode lengths are normalized to 50 steps. The results show that PPO+RND spends roughly 2-4× more time in the noisy-TV room compared to SGCRL. This tendency persists even after the agent has successfully discovered the goal. These results further emphasize that SGCRL demonstrates robustness to irrelevant noise compared to PPO+RND.

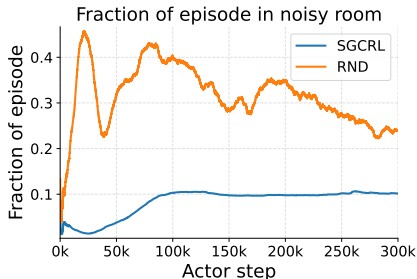

Figure 13: Average fraction of episode time spent in the noisy-TV room.

### D.6.2 IRRELEVANT STATE DIMENSION EXPERIMENT

**Setup.** In this experiment, we again extend the four-room maze with an additional $z$-axis. Unlike the noisy TV setting, the agent now has full control over this dimension and can move freely along $z$ without encountering walls. However, both the initial state and the goal are constrained to lie on the $z = 0$ plane, meaning that optimal behavior does not require exploring the $z$-dimension at all. The purpose of this setup is to test whether SGCRL explores the irrelevant dimension ($z \neq 0$), or whether it efficiently focuses on states relevant to reaching the goal.

**Results.** The results are summarized in Table 2. In this setting, the state space contains 1331 grid cells. After 300k steps, SGCRL explores approximately 40% of this space, while PPO+RND covers about 65%. In absolute terms, PPO+RND visits roughly 332 more cells, which is a substantial difference given the size of the maze. These results again highlight that SGCRL avoids exploring dimensions of the state space that are irrelevant to the goal, in contrast to PPO+RND.

### D.7 SGCRL IS CAPABLE OF REACHING MULTIPLE GOALS

To demonstrate that the SGCRL exploration mechanism extends beyond single-goal settings, we adapt the policy to handle multiple goals simultaneously. Specifically, we extend the SGCRL policy from targeting a single goal to optimizing over a distribution of goals, weighted by their relative desirability. To illustrate this, we present empirical results in the point-maze environment using a tabular setting without neural networks (the same experimental setup as in Section 5). These results show that the single-goal exploration framework naturally generalizes to more complex multi-goal tasks through algorithmic modifications, without requiring neural network architectures.

Formally, consider a task defined by a set of goals

$$\mathcal{G} = \{g_1, g_2, \ldots, g_K\},$$

Here, achieving goal $g_i$ yields reward $r_i$. We modify the action-selection mechanism so that the policy chooses action $a_t$ at state $s_t$ according to the distribution $\text{Softmax}\big(\psi(p(s_t, a_t))^\top \psi_{\text{comb}}\big)$, where the combined goal embedding is defined as

$$\psi_{\text{comb}} := \frac{\sum_{i=1}^K r_i \, \psi(g_i)}{\| \sum_{i=1}^K r_i \, \psi(g_i) \|_2}.$$

The transition model $p_{s_{t+1}} \leftarrow p(s_t, a_t)$ represents the environment dynamics. In our setting, this model is provided to the agent; however, in practical applications it could also be learned. Our purpose here is not to propose the most practical solution, but rather to illustrate—within a simple, tabular model—the computational capabilities of the algorithm. This formulation enables the agent to act according to a reward-weighted combination of goals, rather than committing to a single target.

We evaluate this algorithm in the four-room environment by selecting two goals located at the top-right sub-room, indicated by stars in Figure 14. We assign equal weights $r_1 = r_2 = 0.5$. All representations are initialized with a common component $\mathbf{x} + \varepsilon(s)$, where $\varepsilon(s)$ is state-dependent Gaussian noise. We visualize $\psi$-similarity, i.e., $\psi(s)^\top \psi_{\text{comb}}$, as a heatmap throughout training in Figure 14. Similar to the single-goal setting, we observe that while all representations initially resemble each other (and align

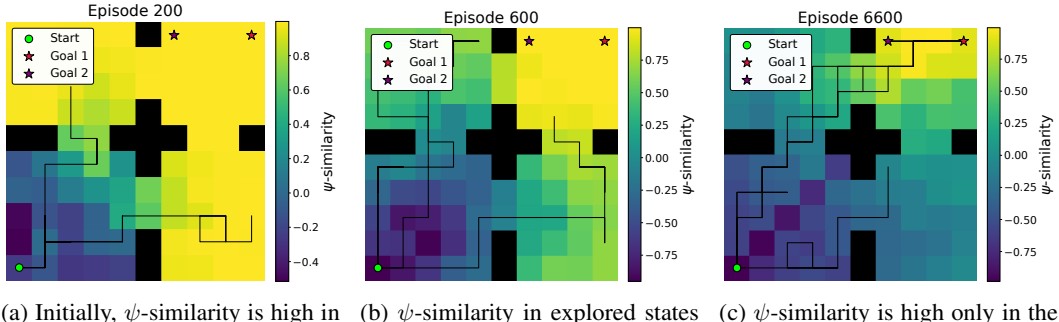

(a) Initially, $\psi$-similarity is high in most states.

(b) $\psi$-similarity in explored states reduces.

(c) $\psi$-similarity is high only in the goals and nearby states.

Figure 14: Representation evolution in the multi-goal task

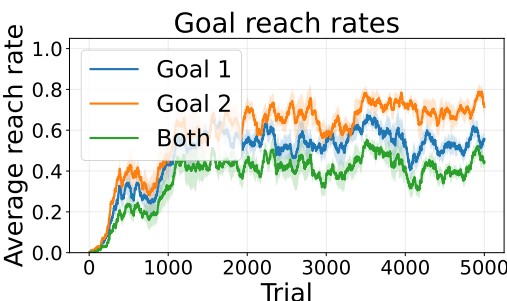

Figure 15: Multi-goal SGCRL maintains a high per-episode reach rate for both goals. Solid lines are agent trajectories

with the combined representation), as the agent explores the environment, they gradually diverge from the common component (reflected by lower $\psi$-similarity values). Eventually, the $\psi$-similarity values stabilize, remaining high in the vicinity of the two goals and low throughout the rest of the environment.

We also plot the average success rate of reaching goal 1, goal 2, and both goals per episode in Figure 15, where the shaded region denotes the standard error across 6 random seeds. The results show that the algorithm manages to reach both goals with nearly equal frequency, and in many episodes it successfully visits both by alternating between them. This experiment demonstrates that SGCRL is capable of handling multi-goal tasks and more complex behaviors, in addition to the simpler single-goal setting. It is worth noting that in the absence of two goals, when only the right hand side goal is chosen, the agent always takes a different path that doesn't pass through the left hand side goal; therefore visiting the left hand side goal in this experiment is not accidental.

## D.8   ABLATING THE REPRESENTATION DIMENSION

We investigate whether changing the representation dimension affects the orthogonality phenomenon predicted by Theorem 2. Although the theorem assumes a high-dimensional representation space, we assess what representation dimension is sufficient in practice for this orthogonality to emerge. Specifically, we analyze the learned representations in a tabular FourRooms environment when the agent explores but fails to find the goal. This scenario is equivalent to selecting a random goal embedding that does not correspond to any actual state in the environment, as in the experimental setup of Figure 4a. We repeat this analysis for a range of representation dimensions.

Our results show that a representation dimension of 4 (or any dimension larger than 4) is sufficient for the representations to become orthogonal when the goal is not found (see Figure 16). However, when we reduce the representation dimension to 2, the orthogonality phenomenon no longer appears (In Figure 16c, the representations lie in a 2D plane and don't move away from the goal representation at

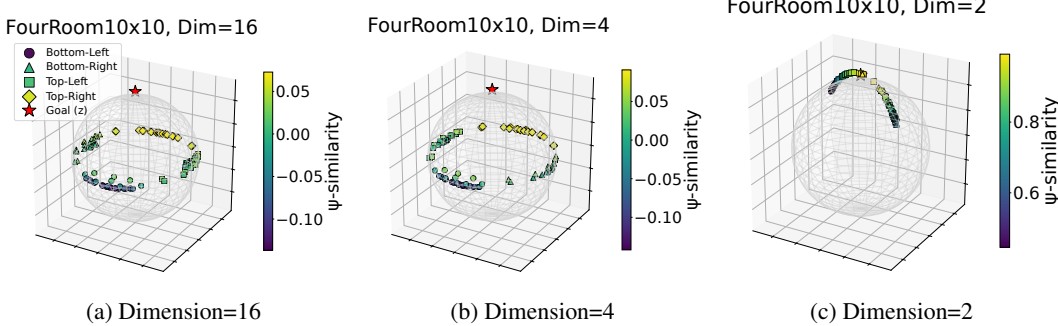

(a) Dimension=16         (b) Dimension=4         (c) Dimension=2

Figure 16: Any representation dimension of four or higher is sufficient to form orthogonal representations in the absence of the goal.

convergence. As a result, the agent collapses into a local minimum and repeatedly visits states near its starting point, since the non-orthogonal representations cannot separate unexplored regions.

## D.9 ABLATING THE REPRESENTATION NORMALIZATION

To investigate the role of normalization in representation formation and in the overall success of the algorithm, we ablate representation normalization in the tabular FourRooms environment.

Theorem 2 suggests that normalization plays a crucial role in the exploration dynamics: because the representation norm is constrained, any component of $\psi(s)$ that was previously parallel to $\psi(g)$ (the "common" component) is forced into the orthogonal subspace. Without normalization, however, the representations can continue to encode useful information without suppressing this parallel component.

We evaluate this hypothesis by removing representation normalization in the tabular setting. We find that normalization is indeed essential: in the $10 \times 10$ FourRooms environment with an episode length of 30, the non-normalized representations achieve *zero* success. It is worth noting that with a longer episode length of 100, the non-normalized variant can eventually solve the task, but we report the more challenging 30-step setting to evaluate performance under stress conditions.

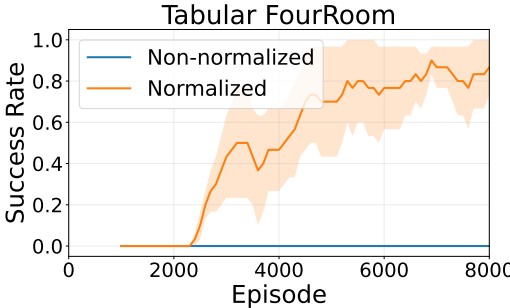

Figure 17: In the tabular experiment, representation normalization plays a very important role in success.

## D.10 ABLATING THE CRITIC LOSS

We now consider the case where the critic is learned via temporal-difference (TD) learning. In particular, suppose the discounted successor measure admits the low-rank form

$$p_\gamma^\pi(s_f \mid s, a) = \phi(s, a)^\top \psi(s_f).$$

To update the representations using TD, we minimize the squared TD error

$$\mathcal{L}_{\phi,\psi}^{\text{TD}} = \left( \phi(s, a)^\top \psi(s_f) - \left[ (1 - \gamma)\mathbf{1}[s = s_f] + \gamma\,\phi(s', a')^\top \psi(s_f) \right] \right)^2, \tag{23}$$

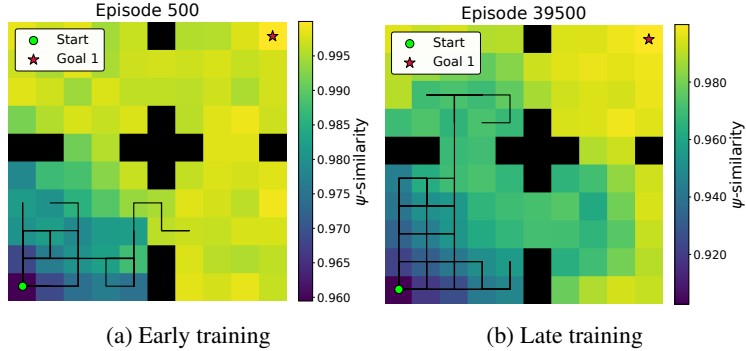

(a) Early training (b) Late training

Figure 18: A TD-style critic produces the same qualitative pattern as CRL: the $\psi$-similarity of visited states decreases over training. However, despite this representational effect, the agent consistently fails to reach the goal.

where $s_f$ is sampled at a geometrically distributed time offset from $t$, $(s', a')$ is the next transition in the replay buffer, and $a' \sim \pi(\cdot \mid s')$.

Let the TD error be

$$\delta := \phi(s, a)^\top \psi(s_f) - \left[(1 - \gamma)\mathbf{1}[s = s_f] + \gamma\,\phi(s', a')^\top \psi(s_f)\right].$$

The gradient-descent updates are then

$$\phi^{(t)}(s, a) = \phi^{(t-1)}(s, a) - 2\eta\,\psi(s_f)\,\delta,$$

$$\psi^{(t)}(s_f) = \psi^{(t-1)}(s_f) - 2\eta\,\phi(s, a)\,\delta,$$

followed by normalization of the representations. This TD-based procedure can be viewed as an alternative to the contrastive CRL objective for learning a low-rank approximation of the successor representation.

We implemented this TD-learning variant in the tabular FourRooms environment. While the learned representations exhibit qualitatively similar structure to those obtained with the contrastive method—states visited frequently during exploration acquire lower $\psi$-similarity, the agent consistently fails to reach the goal across five random seeds, refer to Figure 18.

### D.11 ANALYZING THE GOAL SIMILARITY IN THE CONTINUOUS SETTING

In this section, we examine whether the orthogonality result of Theorem 2 continues to hold in the continuous setting. The theorem assumes that the representations of non-goal states are updated while $\psi(g)$ remains fixed, modeling the situation where the agent explores regions of the state space without encountering the goal. This assumption is exact in the tabular case, where updates are local. However, in practice, when representations are produced by a shared neural encoder, updating states far from the goal can still modify $\psi(g)$ indirectly.

To test whether the orthogonality effect persists despite this coupling, we run SGCRL in the continuous FourRooms environment and choose a goal embedding corresponding to an out-of-bounds coordinate, $g = (20, 20)$, ensuring that the agent never observes the true goal state. We then track the evolution of goal similarity throughout training across multiple seeds. See Figure 19. Consistently, we observe the same pattern as in the tabular setting: representations initially have high $\psi$-similarity to the goal but gradually decay toward values near zero, reflecting an approximate orthogonalization. Although the similarity does not reach exactly zero, it becomes close, indicating that the theoretical prediction remains a good approximation even with a shared encoder.

### D.12 SAFETY BASELINE EXPERIMENTS

To better contextualize our safety experiments, we evaluate a few baseline safe RL algorithms from the Safe Policy Optimization algorithm benchmark (Ray et al., 2019) on the FourRooms safety

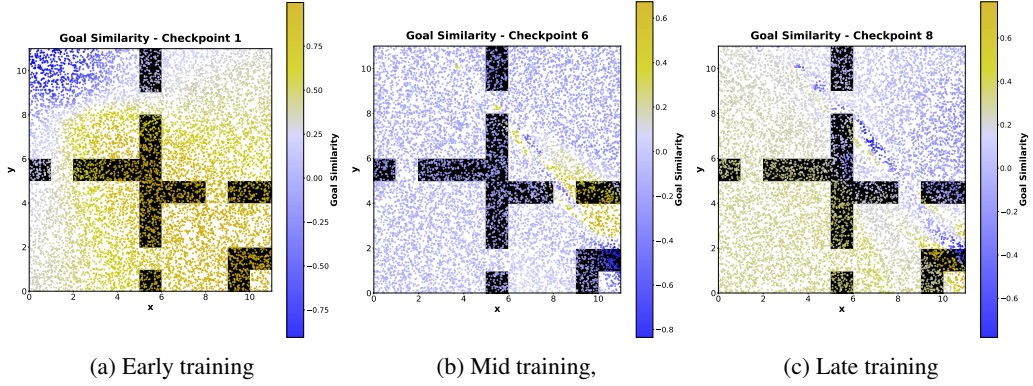

(a) Early training    (b) Mid training,    (c) Late training

Figure 19: Even with a shared neural encoder, updating representations in regions far from the goal drives the representations of visited states to become approximately orthogonal to the goal embedding. Starting point is at the top left corner and the impossible goal coordinate is $(20, 20)$.

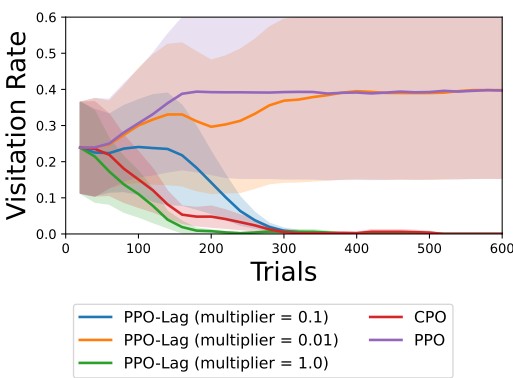

Figure 20: Comparison of safety violation rate for baseline safe RL algorithms

environment. In this environment, the states in the bottom left room are considered "unsafe". We evaluate the following baselines:

- PPO: Proximal Policy Optimization (PPO) (Schulman et al., 2017) serves as an unconstrained baseline.

- PPO-LAG: PPO-Lagrangian (Ray et al., 2019) uses an adaptive penalty coefficient on the PPO objective to enforce safety constraints. This method equivalently solves the unconstrained max-min optimization problem

$$\max_{\theta} \min_{\lambda \geq 0} \mathcal{L}(\theta, \lambda) = f(\theta) - \lambda g(\theta), \tag{24}$$

  where $f(\theta)$ is the objective and $g(\theta) \leq 0$ is the constraint. We set the cost-limit hyperparameter to $1.0$ (no tolerance to safety violations). We tune over values for the initial penalty coefficient.

- CPO: Constrained Policy Optimization (Achiam et al., 2017) enforces constraints by solving trust region optimization problems analytically at each policy update. We set the cost-limit hyperparameter to $1.0$ (no tolerance to safety violations).

Figure 20 shows the safety violation rate of these baselines. We do not plot SGCRL in this comparison because standard SGCRL converges at a much slower rate than these lightweight, online baselines (Fig. 5c).

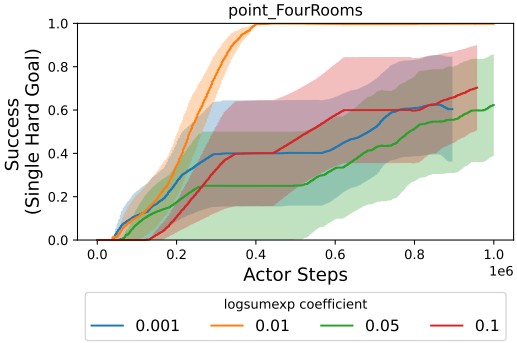

Figure 21: Role of the logSumExp term in the success rate when using the InfoNCE forward loss.

## D.13 ABLATING FORWARD VS BACKWARD INFONCE LOSS AND THE ROLE OF THE LOGSUMEXP TERM

While the main experiments use the InfoNCE backward loss to train the critic (Eq. 1), in this section we include an ablation in which we also evaluate the InfoNCE forward loss (Equation 25), which also contains an additional log-sum-exp term.

$$\max_{\phi,\psi} \mathbb{E}_{\substack{(s_i,a_i)\sim p_{\mathcal{D}}(s,a) \\ s_f^{(i)}\sim p_\gamma^\pi(\cdot|s_i,a_i) \\ i=1,\dots,N}} \left[ \frac{1}{N} \sum_{i=1}^N \log \frac{\exp\big(\phi(s_i,a_i)^\top \psi(s_f^{(i)})\big)}{\sum_{j=1}^N \exp\big(\phi(s_i,a_i)^\top \psi(s_f^{(j)})\big)} - \alpha \log\big(\sum_{j=1}^N \exp \phi(s_i,a_i)^\top \psi(s_f^{(j)})\big)^2 \right],$$

(25)

Our goal is to analyze the role of this log-sum-exp factor in shaping the learned representations and in determining the agent's success rate. We sweep the forward-loss temperature parameter using $\alpha \in 0.001, 0.01, 0.05, 0.1$ in the tabular four-room environment and observe no meaningful differences in representation dynamics. In particular, the evolution of $\psi-$ similarity during exploration—when the agent visits states far from the goal—exhibits the same orthogonality pattern as with the backward loss.

In the continuous setting, however, varying $\alpha$ does affect the agent's rate of convergence. We include these results in Figure 21. The representation patterns appear similar across difference values of $\alpha$ (Figure 22).

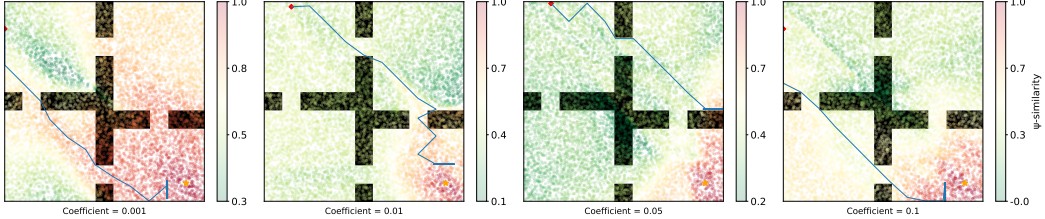

Figure 22: Representations during training for different logSumExp coefficient values

## D.14 ADDITIONAL EXPERIMENTS ON THE IMPORTANCE OF SINGLE-GOAL DATA COLLECTION

To further test the hypothesis that exploration with a single hard-goal representation is responsible for the orthogonality effect described above, we conduct an additional experiment that mirrors the setup of Appendix D.5. The only change is in the data-collection policy: instead of comparing single-goal exploration to a completely random-goal policy, we now compare it to a family of multi-goal exploration strategies that interpolate between these two extremes.

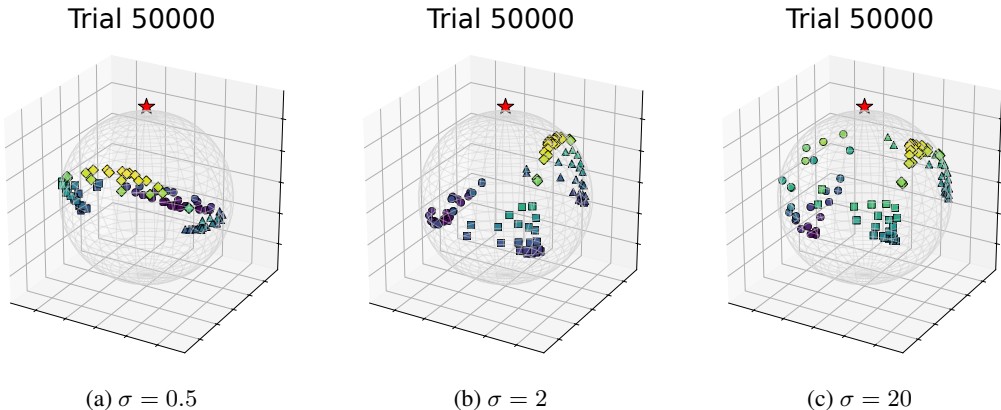

| (a) $\sigma = 0.5$ | (b) $\sigma = 2$ | (c) $\sigma = 20$ |

Figure 23: As the data-collection goal representation deviates further from $\psi(g)$ (larger $\sigma$), the learned representations are less strongly pushed toward the orthogonal subspace.

Concretely, in each episode we sample a goal embedding

$$\mathbf{y}_e \;=\; \psi(g) + \sigma\,\boldsymbol{\epsilon}, \qquad \boldsymbol{\epsilon} \sim \mathcal{N}(0, I_d),$$

and use $\mathbf{y}_e$ in place of $\psi(g)$ in the multi-goal exploration policy. The scalar $\sigma$ controls how tightly the sampled goals are concentrated around the hard goal representation $\psi(g)$: small values of $\sigma$ correspond to goals tightly clustered around $\psi(g)$ (approaching the single-goal regime), while large values of $\sigma$ recover a widely spread goal distribution similar to the random-goal setting. We study the evolution of the representations for $\sigma \in \{0.5, 2, 20\}$.

Our expectation is that, when $\sigma$ is small, the induced data collection remains close to single-goal exploration, leading to "clean" representation formation in which all frequently visited, non-goal regions are driven far from $\psi(g)$ and thus enjoy the same orthogonality effect. As $\sigma$ increases, the data collection becomes less aligned with the hard goal, and we anticipate that this degrades representation quality: many visited states retain high similarity to $\psi(g)$, preventing the agent from escaping already explored regions and ultimately reducing exploratory coverage. The representation evolution under these varying data-collection strategies is summarized in Figure 23.

**Multiple reachable goal data collection**    While the previous experiment studied representation formation under an *unreachable* goal, we now consider a setting in which the goal is *reachable* and compare single-goal data collection with multi-goal data collection using two and four goals.

In the two-goal setup, the exploration goal is sampled uniformly at random from $\{g_1 = (9, 9), g_2 = (6, 6)\}$ at the beginning of each episode; And in the four-goal setup, the exploration goal is sampled uniformly from $g_1, g_2$ or $g_3 = (0, 9), g_4 = (9, 0)$. In all cases, the evaluation goal — used to measure success — is fixed to the hardest goal, $(9, 9)$.

Figure 24 shows the resulting success rates. The results clearly indicate that exploration with a single goal leads to more effective representations and higher goal-reaching performance than distributing exploration across multiple goals. This finding is consistent with the results of Liu et al. (2025).

Beyond success rates we also visualize the $\psi$-similarity across the three data-collection regimes and the different goals. Note that in this realm since the goal is reachable, we do not expect the representations to become fully orthogonal to the goal because Theorem 2 only predicts exact orthogonality when the goal is unreachable. But we expect states that are far from the goal should exhibit grow low $\psi$-similarity over time, while states closer to the goal — especially those along typical paths to the goal — should retain high similarity.

With this intuition in mind, we investigate two key questions:

RQ1 When visualizing $\psi(s)^\top \psi(g_i)$ for each of the four goals $g_i$, $i \in \{1, 2, 3, 4\}$, do we observe the coherent structure in which states far from each goal exhibit low similarity, while states near the goal and along the path to it exhibit high similarity for all 4 goals at the same time?

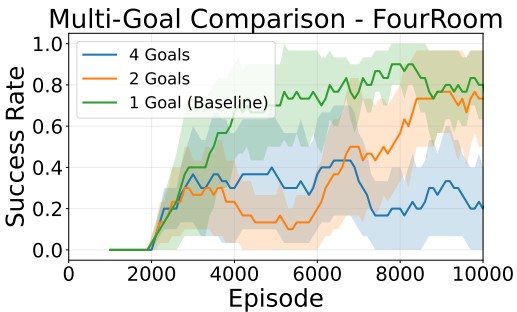

Figure 24: Success rate is the highest when we only collect data with the single hard goal. Tabular experiment in the FourRoom environment.

RQ2 When focusing solely on the hardest goal $g_1 = (9, 9)$, do the heatmaps of $\psi(s)^\top \psi(g_1)$ differ depending on whether the data was collected using only $g_1$, or using two or four exploration goals? Specifically, are the representations learned through single-goal data collection superior?

We first address RQ1. The similarity maps $\psi(s)^\top \psi(g)$ for the four different goals in the four-goal data-collection experiment are shown in Figure 25. While we do observe the general trend that states far from each goal have low similarity and states closer to the goal have higher similarity for all four goals, the structure is noticeably less precise than when data is collected using a single goal.

For instance, Figure 26a shows the representation when data is collected using only one goal, where we observe a clean and smooth gradient of similarity toward the goal. In contrast, in the four-goal setting (Figure 25a), the gradients are much less structured, and the model does not clearly distinguish between $g_1$ and $g_2$.

An even clearer example is shown in Figure 25d, where almost the entire left side of the maze exhibits high similarity with $g_4$, even though many of these states are actually far from $g_4$. This indicates that when exploration is done with multiple goals, the learned representations become less sharply aligned with the true spatial structure of the environment.

We now address RQ2. Refer to Figure 26 for a comparison of $\psi(s)^\top \psi(g_1)$ across different data-collection strategies. We again see that, for multi-goal data collection strategies, although overall states far from the goal have low similarity, there are still many states at distance 3–4 from the goal that exhibit high similarity and those areas are where the agent gets stuck.

We hypothesize that this happens because the path from the starting point to $g_1 = (9, 9)$ and $g_2 = (6, 6)$ is partially shared. Since the dataset contains trajectories that follow the same path but sometimes terminate at $g_1$ and sometimes at $g_2$, the representation of $g_2$ becomes similar to that of $g_1$. As a result, the agent can get "stuck" at $g_2$ even when it is commanded to reach $g_1$. In the single-goal data-collection setting, this failure mode is avoided: the agent is always commanded to go to the same goal, so irrelevant states such as $g_2$ do not acquire $g_1$-like representations that would cause the policy to stall there.

### D.15 Additional Safety Experiment Results

Here, we provide additional plots to demonstrate that the safety intervention experiments (Fig. 5c) hold when the intervention is conducted in either the top-right or bottom-left room of the FourRooms environment.

## E The Use of Large Language Models

We used LLMs while implementing experiments to generate boilerplate code, debug errors, and plot results. We wrote the paper manuscript manually but used LLMs to help edit writing for clarity and grammar.

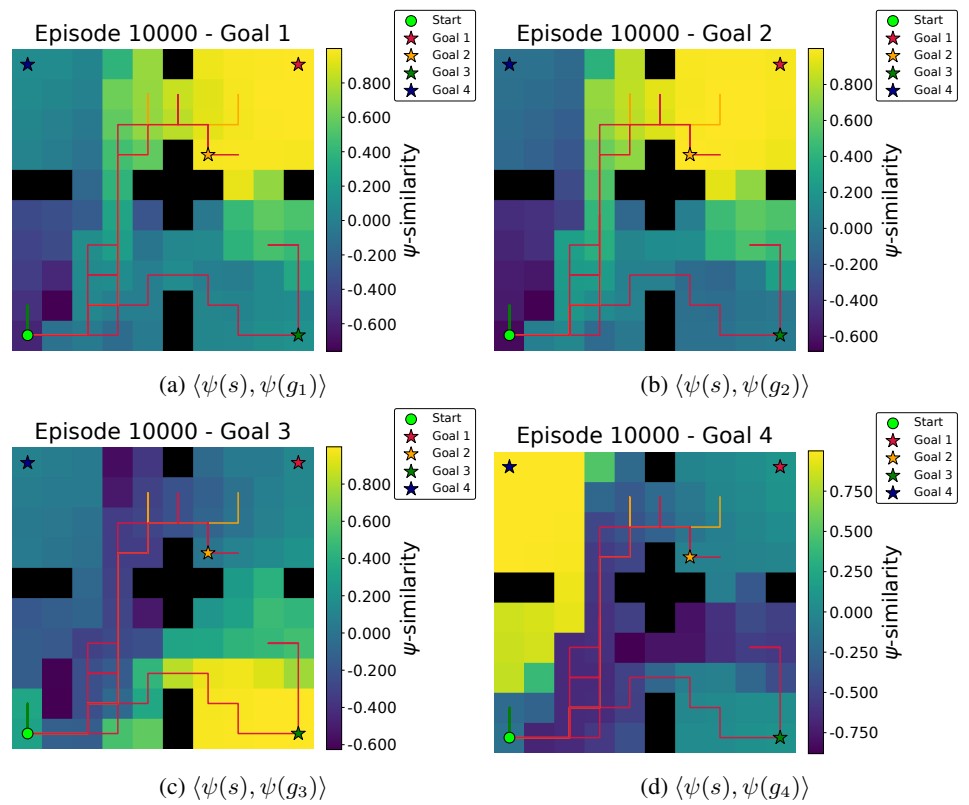

Figure 25: Data collection with 4 goals. Heatmaps shows the goal similarities with respect to each of the four goals. The trajectory over 5 episodes is shown, with line colors indicating the commanded goal.

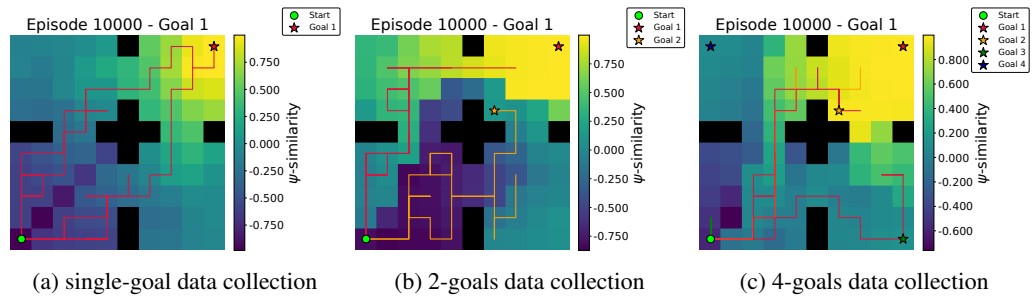

Figure 26: Comparison of representations with respect to $g_1$ under single-goal, two-goal, and four-goal data collection.

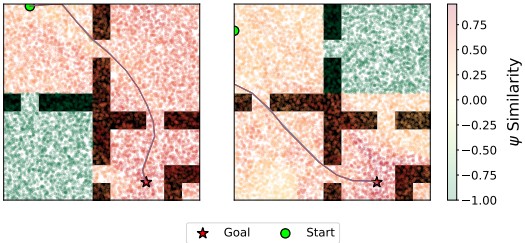

Figure 27: Modifying the representations to be dissimilar to the goal improves controllability of the agent's behavior.

