# OpenReview forum: "Demystifying The Mechanisms Behind Emergent Exploration in Goal-Conditioned RL"
_ICLR.cc/2026/Conference — ICLR 2026 Poster_

### Official Review · Reviewer_2Td2 · 2025-10-31

**Soundness:** 3
**Presentation:** 4
**Contribution:** 3
**Rating:** 6
**Confidence:** 4

**Summary:**

This paper aims to demystify the mechanisms behind emergent exploration in Single-Goal Contrastive Reinforcement Learning (SGCRL). The authors combine theoretical analysis and controlled experiments both in continuous and tabular settings to argue that SGCRL implicitly maximizes a representation-based reward defined by goal similarity. This implicit reward gives rise to an automatic exploration–exploitation curriculum: states along unsuccessful trajectories gradually lose goal similarity (discouraging revisits), while those along successful paths gain similarity (encouraging exploitation). The authors show that these dynamics persist even without neural network function approximation, suggesting they arise from the low-rank structure of contrastive representations rather than network inductive biases. They further propose that understanding these dynamics enables safety-aware exploration by manipulating representations.

**Strengths:**

* The paper provides a clear and well-structured theoretical account of why SGCRL exhibits emergent exploration despite lacking explicit rewards.
* I think the use of tabular environments to isolate the contrastive representation mechanism is elegant and persuasive.
* The paper is well written, and the figures effectively communicate the representation dynamics.
- The demonstration that altering representation geometry can control exploration pathways is an intriguing direction for safe RL.

**Weaknesses:**

- I feel the the proposed interpretation, exploration as maximisation of a representation-based implicit reward, is closely aligned with existing frameworks such as successor representations, potential-based reward shaping, and eigenoptions (e.g. (Machado et al)). The connection to these earlier ideas is not fully acknowledged, giving the impression of a new theory rather than a reinterpretation.
- The framing suggests spontaneous behaviour arising from complex dynamics, but the mechanism can be straightforwardly understood as representation-induced reward shaping.
- The intervention experiments are promising but not systematically evaluated or compared to existing safety-aware exploration methods.

**Questions:**

1. How does the proposed “implicit reward” interpretation differ formally from the successor feature formulation of value functions?

2. Would the same dynamics appear if the critic used a non-contrastive predictive loss (e.g., temporal difference learning or policy gradient)? One could argue that with the policy gradient once the agent is near convergence a similar patterns appears, i.e., that the described phenomenon in the paper might not be exclusive to the contrastive approach

3. How sensitive are the observed behaviours to representation normalisation or embedding dimensionality?  I would assume normalisation/compression plays a big role in shaping the geometry of the reward function.

---

> ### Author Response · Authors · 2025-11-20
>
> Dear Reviewer,
>
> Thank you for your thoughtful and detailed feedback. It seems that the main concerns related to the framing of our contribution in context of previous work. To address these concerns and other comments, we have revised our **related works section and section 4.1** of the paper and **added new experiments in Appendices D.8, D.9, D.10, D.12**. All newly added content in the paper and appendix is highlighted in yellow. **Do the new analysis and experiments address the reviewer's concerns about the paper?**
>
> ---
>
> > The connection to these earlier ideas is not fully acknowledged
>
> Thank you for the pointers to relevant works! We added a new paragraph at the end of Section 4.1 clarifying the relationship between our results and the prior literature on successor representations (SR) and successor features. Although our theoretical analysis identifies a shaped reward and a structure reminiscent of successor features, we want to emphasize that the algorithm we study is **not explicitly designed** to maximize a discounted sum of a shaped reward, nor does it include any architectural component that enforces an exploration-inducing curriculum on that reward. Instead, the connection to SR and the resulting exploration guarantees arise *implicitly* from the fixed-point analysis of the InfoNCE objective.
>
> Moreover, prior SR work does not study SR from the perspective of *exploration*. To the best of our knowledge, our work is the first to show how, under contrastive RL, both (i) a shaped reward and (ii) successor-like features arise implicitly and **jointly evolve to induce exploration**. We highlight these distinctions explicitly in the added paragraph in Section 4.1.
>
> ---
>
> > The framing suggests spontaneous behaviour arising from complex dynamics, but the mechanism can be straightforwardly understood as representation-induced reward shaping
>
> While our analysis produces a simple and interpretable explanation, we view this simplicity as a strength rather than a weakness. Importantly, none of the exploration-driving mechanisms we identify are explicitly built into the algorithm. For instance, SGCRL is defined purely as maximizing the probability of reaching the goal under its learned representations—its original formulation includes no notion of dynamic reward shaping or any hand-crafted exploration incentive.
>
> Despite its apparent simplicity, our explanation provides insight into how SGCRL can solve tasks as challenging as the 5-disk Tower of Hanoi without relying on neural-network generalization. Moreover, recent work (e.g., Nimonkar, 2025) shows that the same algorithm can form specialized, emergent skills in multi-agent settings. We aim to frame our theoretical analysis to highlight how seemingly complex behavior arises from *simple* underlying dynamics. Finally, "spontaneous behavior arising from complex dynamics" was the prevailing wisdom and explanation behind the success of SGCRL beforehand and substituting that prior explanation with a "straightforwardly understood" mechanism is the main contribution of our work.
>
> ---
>
> > existing safety-aware exploration methods.
>
> We have added an analysis of safe RL baselines on our FourRooms safety task in **Appendix D.12**. We use safe RL algorithms from the Safe Policy Optimization Benchmark (Ray et al. 2019): Proximal Policy Optimization (PPO), PPO + Langrangian Constraint (PPO-LAG), and Constrained Policy Optimization (CPO).
>
> We do note that these online baselines converge an order of magnitude faster than standard (continuous) SGCRL, which makes use of a replay buffer and larger networks. Therefore, we do not claim that SGCRL in its current form is an efficient, safe RL algorithm, but highlight this as an exciting avenue for future work. Nonetheless, these experiments provide a way to experimentally test hypotheses drawn from our theory. Thus, the success of the experiments at safety (admittedly with worse sample complexity than SOTA safety methods) provides empirical support of our theory.
>
> ---
>
> > Would the same dynamics appear if the critic used a non-contrastive predictive loss
>
> We added a new experiment in **Appendix D.10**, where we examine how the algorithm behaves when the critic is trained with a TD loss instead of the InfoNCE loss. We additionally include tabular experiments demonstrating that, the TD-based variant fails to solve the task, though the representation evolution resembles that of SGCRL.
>
>
>
> References:
>
> Nimonkar, C., Shah, S., Ji, C., & Eysenbach, B. (2025). Self-Supervised Goal-Reaching Results in Multi-Agent Cooperation and Exploration.
> Ray, A., Achiam, J., & Amodei, D. (2019). Benchmarking safe exploration in deep reinforcement learning.

---

> > ### Author Response · Authors · 2025-11-20
> >
> > As a continuation of the previous comment, we also address the reviewers’ final concern below
> > ---
> > > How sensitive are the observed behaviours to representation normalisation or embedding dimensionality
> >
> > We added new experiments in **Appendix D.8** and **D.9** to ablate both representation dimensionality and normalization. Our experiments show:
> >
> > - Even very low-dimensional representations (e.g., 4D) exhibit the same qualitative alignment/orthogonality dynamics.
> > - However, **normalization plays a crucial role** in the success of the algorithm.
> > ---

---

### Official Review · Reviewer_GW6a · 2025-10-31

**Soundness:** 3
**Presentation:** 4
**Contribution:** 3
**Rating:** 6
**Confidence:** 3

**Summary:**

This work analyses the origins of implicit exploration behaviors observed in Single-Goal Contrastive Reinforcement Learning (SGCRL), a model that actively aligns state-action representations with the representations of a single goal. The paper presents a theoretical and experimental analysis of the dynamics of state and goal representations learned by the model. A long list of experiments characterizes the desiderata to make SGCRL induce a form of decaying optimism that boost exploration.

The paper is well-written and the work provides a sound and novel analysis that helps understand the emergence of implicit exploration behaviors in a RL setting. Thus, I recommend acceptance.


I'd be happy to further increase my score:
Section 4.2 kicks in an open door. Given the infoNCE loss function, it is not surprising that the similarity between phi(g) and phi(s) rules the behavior. The paper decides to focus on Single-Goal RL and shows in Appendix that uniform goal sampling does not work. The experimental section could provide an analysis of how the number of fixed goals, along with their goal-conditioned policy, affect the representations and the exploration behaviors. Single-Goal RL is a limited setting, extending the analysis may bring more general results or significant insights for future work.


Minor comments:
- Figure 5a), please, bring the star forward.
- The benefits of intervening on the representations rather than with a direct reward is unclear.

**Strengths:**

The paper is well-written and the work provides a sound and novel analysis that helps understand the emergence of implicit exploration behaviors in a RL setting.

**Weaknesses:**

The analysis focuses on a constrained setting, Single-Goal contrastive learning.

**Questions:**

Please, see above.

---

> ### Author Response · Authors · 2025-11-20
>
> Dear Reviewer,
>
>
>
> Thank you for the thoughtful and detailed feedback. It seems like the main concern relates to the theoretical contribution and the practicality of the single-goal setting. We have revised the paper accordingly and added **new experiments to Appendix D.14**. All newly added content in the paper and appendix is highlighted in yellow. **Do the new analysis and experiments address the reviewer's concerns about the paper?**
>
>
>
> ---
>
>
>
> > it is not surprising that the similarity between phi(g) and phi(s) rules the behavior
>
> We would like to highlight the significance of our theoretical contributions. Although some of the results may appear intuitive in hindsight, we believe they provide two key advances:
>
> - **First**, we show that—under Assumption 1 (for which we provide theoretical justification in Appendix B.2.1 and B.2.2)—the quantity $\psi$ behaves as a *shaped reward* and $\phi$ corresponds to *successor features*. To our knowledge, this is the first work to formally establish this connection for contrastive RL and to link it directly to exploration.
>
> - **Second**, our theory explains *why*, under normalized representations, this emergent shaped reward induces systematic exploration. This offers a principled understanding of the exploration behavior observed empirically.
>
> ---
> > could provide an analysis of how the number of fixed goals, along with their goal-conditioned policy, affect the representations and the exploration behaviors.
> >
> We hypothesize that the key factor influencing representation formation is how closely the exploration-time goal representation resembles the true hard-goal representation $\psi(g)$. To assess the role of the exploration goal in shaping the learned representations, we conducted an additional set of experiments. **Appendix D.14** presents these results, examining representation evolution across a spectrum of goal-selection strategies that range from being tightly aligned with $\psi(g)$ to increasingly distant from it.
>
>
> ---
>
> > Single-Goal RL is a limited setting
>
> We have revised our related works section to address this concern. To clarify the scope of our setting, we distinguish between **test-time** and **training-time** goals.
>
> In regards to test-time goals, our analysis and experiments focus on tasks which have a single hard goal at test time. These goals are often semantically meaningful (e.g. having all disks on the 3rd peg in Towers of Hanoi). Although our main analysis focuses on single-goal reaching tasks, we perform some experiments showing how the single-goal framework can be adapted to achieve *a distribution of test-time goals* while retaining the benefits of single-goal representation shaping (**Appendix D.7**).
>
> In regards to train-time goals, we want to clarify that SGCRL (and CRL more generally) is a HER-style goal-conditioned algorithm because it performs learning updates using future states relabeled as goals. The *single-goal* property refers to the fact that SGCRL is commanded only a single-goal (the test-time goal) during data collection. Previous work (Liu et al. 2025) showed that simply commanding this single-goal outperforms other HER-style baselines as well as a CRL baseline that employs a human-designed curriculum of data collection subgoals. Therefore, our contribution is to explain the surprising result of how *single-goal* data collection alone produces strong exploration. To make this distinction explicit, we added a third paragraph to the Related Work section clarifying our scope and positioning our contribution relative to prior multi-goal approaches. Moreover, we want to highlight **Appendix D.5**, which shows that *single-goal data collection shapes representations more effectively* than collecting data with a multi-goal distribution. This analysis provides insight into why SGCRL outperforms CRL with a manually-designed goal curriculum.
>
>
>
> ---
> > benefits of intervening on the representations rather than with a direct reward is unclear.
>
> One major advantage of intervening on representations rather than rewards is that it allows us to improve controllability for algorithms that don't use rewards (such as SGCRL). SGCRL is fully unsupervised: both the actor and critic operate purely on representation similarity and do not make use of explicit reward functions. As a result, there is no clear mechanism for incorporating such a reward into either the actor or the critic, since their updates are defined entirely in the representation space. In contrast, intervening directly on the learned representations aligns naturally with the algorithm’s structure and allows us to study and control its behavior without altering its fundamental training objective. Of course, in problem settings with reward supervision, intervening on rewards could be more appropriate.
>
> ---

---

> > ### Comment · Reviewer_GW6a · 2025-11-24
> > **response**
> >
> > I thank the authors for their response and clarifications.
> >
> > > We hypothesize that the key factor influencing representation formation is how closely the exploration-time goal representation resembles the true hard-goal representation $\psi(g)$. To assess the role of the exploration goal in shaping the learned representations, we conducted an additional set of experiments. Appendix D.14 presents these results, examining representation evolution across a spectrum of goal-selection strategies that range from being tightly aligned with $\psi(g)$ to increasingly distant from it.
> >
> > This paper claims that "single goal" works much better than "multi-goal". This work analyses the success of single-goal CRL, but this does not really explain why multi-goal CRL does not work. An analysis, even experimental, in the main paper would significantly strengthen this work. Appendix D.14 is partially what I was looking for. However, I have several comments there.
> > - Testing discrete number of goals (2, 4, 8), with or without orthogonality between them could complete this analysis. With the Gaussian distribution studied in Appendix D.14, the rank of the space orthogonal to all goals is drastically reduced, which may impair representation learning. Is that why it does not work ? Using a discrete number of controlled goals allows one to investigate this.
> > - The upper part of InfoNCE, with a uniform distribution of state-actions, could quantitatively reflect the visualizations provided in Figure 23. Such a metric would facilitate analyzing how different distributions of goals impact the representation learning process.
> > - For a discussion: in self-supervised visual learning, several works already highlight that InfoNCE-like loss functions learn a low-rank set of input embeddings (e.g [1,2]). That is why this research community extracts the high-rank intermediate representations rather than the final embeddings for evaluating their models. Can the authors discuss their results with respect to these works ? Is the low rank a problem or an advantage regarding SGCRL ? How would multi-goal CRL behave with an embedding of higher rank ?
> >
> > [1] Gupta K, Ajanthan T, Hengel Avd, Gould S. Understanding and improving the role of projection head
> > in self-supervised learning. arXiv preprint arXiv:221211491. 2022.
> > [2] Jing L, Vincent P, LeCun Y, Tian Y. Understanding Dimensional Collapse in Contrastive Self-
> > supervised Learning. In: International Conference on Learning Representations;. .

---

> ### Author Response · Authors · 2025-11-29
> **Author Response**
>
> Thank you for your thoughtful response and additional suggestions! We have responded to the followup with an additional experiment (Appendix B.14) and additional discussion in Section 6 of the paper.
>
> ---
>
> > Testing discrete number of goals (2, 4, 8), with or without orthogonality between them could complete this analysis. With the Gaussian distribution studied in Appendix D.14, the rank of the space orthogonal to all goals is drastically reduced, which may impair representation learning. Is that why it does not work ? Using a discrete number of controlled goals allows one to investigate this.
>
> We clarify that our main contribution is not to show that single-goal exploration works better than multi-goal exploration — this result already appears in Liu et al. (2025). Instead, our primary contribution is to provide a **mechanistic explanation** for why single-goal exploration works at all.
>
> However, we agree that understanding representation behavior under multi-goal data-collection regimes is valuable. To investigate this, we conducted a new experiment (**Appendix D.14**) in which we collect data using 1, 2, or 4 reachable exploration goals and analyze both the success rate and the evolution of representations. Two important considerations about this experiment:
> - In this experiment as well as all of our other tabular experiments, all states are initialized with identical representations; Because we want to capture the fact that the algorithm initially cannot distinguish between states. And the representations evolve during training through contrastive learning. Because of this, we cannot directly construct the exact experimental setup the reviewer suggested (i.e., enforcing goal representations to be orthogonal or not). Nevertheless, our experiments do clearly reveal how representations evolve differently under single-goal versus multi-goal data collection, which is precisely the question the reviewer asked.
>
> - In this experiment since the goals are reachable, the assumptions of Theorem 2 do not hold, and we do not expect exact orthogonality (as opposed to the experiment setup of Appendix D5). Nevertheless, we still expect states far from the goal to gradually develop low similarity to the goal over time (though not necessarily zero).
>
> We find that collecting data with multiple goals leads to much lower success rates. In the single-goal setting, goal similarity forms a clean gradient: states become less similar in the representation space as their temporal distance from the goal increases. This structure breaks down with multiple goal data collection. Our hypothesis is that, when doing multi-goal data collection, the agent often visits states irrelevant to the hardest goal, yet some of these states still increase in hardest-goal similarity because they share early parts of the trajectory (common predecessors) with the hardest goal. As a result, the agent gets stuck focusing on misleading states far from the true goal. In contrast, single-goal data collection avoids these distractions, because it is never commanded to reach any states that are irrelevant to the hardest goal.
>
> For a more detailed discussion and additional visualizations, please refer to the final part of Appendix D.14.
>
> > The upper part of InfoNCE, with a uniform distribution of state-actions, could quantitatively reflect the visualizations provided in Figure 23. Such a metric would facilitate analyzing how different distributions of goals impact the representation learning process.
>
> Thank you for the suggestion! Our interpretation is this proposes a way of measuring alignment loss for the plots in Figure 23, though we are not sure if the suggestion is referring to a way to measure rank collapse (per the discussion below), which may be better measured using the "bottom part" of the InfoNCE loss.

---

> > ### Author Response · Authors · 2025-11-29
> > **Author Response (continued)**
> >
> > > For a discussion: in self-supervised visual learning, several works already highlight that InfoNCE-like loss functions learn a low-rank set of input embeddings (e.g [1,2]). That is why this research community extracts the high-rank intermediate representations rather than the final embeddings for evaluating their models. Can the authors discuss their results with respect to these works ? Is the low rank a problem or an advantage regarding SGCRL ? How would multi-goal CRL behave with an embedding of higher rank ?
> >
> > Thank you for the pointers to relevant work on mitigating representation collapse with InfoNCE loss functions. Our understanding is that Gupta et al. and Jing et al. demonstrate that in self-supervised visual learning, InfoNCE-like losses learn low-rank representations, which is considered detrimental to downstream classification tasks due to loss of representational capacity. Our paper studies a different problem setting, which helps to explain why our work comes to different conclusions about low-rank representations. We explain this in details below, and include the discussion in Section 6 of the revised paper.
> >
> > For SGCRL, limited representational capacity is not a weakness, but rather a necessary component of the method's success. More specifically, the mechanism that allows SGCRL to prune the search space relies on the geometric constraints imposed by low-dimensional, normalized embeddings. Before the goal is found, the InfoNCE loss drives the representations of states along unsuccessful trajectories to become orthogonal to the goal embedding (Fig. 4), a form of dimensional collapse that incentivizes exploration towards unvisited states. This happens because the InfoNCE objective, when operating on normalized embeddings of limited capacity, suppresses shared components (e.g. psi(g)) that do not help learn temporal differences (Thm. 2). Indeed, we do find that for tabular settings, the effective dimension of the representations converge to ~3-4. Therefore, we propose that the low rank property is a necessary condition for efficient emergent exploration.
> >
> > To further investigate the role of the representations, we conducted an ablation where we replaced the vectorized (16-dim) representations with a $|\mathcal{S}|\times|\mathcal{S}|$ lookup table of state-goal similarity, learned with the InfoNCE loss. We find that using this lookup table requires approximately $100\times$ more samples to solve the FourRooms task (**Appendix D.3**). The results of this ablation further suggest that the low-dimensional, vectorized representation structure is a necessary component for efficient exploration of SGCRL. These results also suggest that a higher-rank embedding would likely not improve exploration efficiency for SGCRL and Multi-goal CRL. For both algorithms the primary exploration mechanism relies on the limited representational budget of normalized and low-dimensional representations. The primary practical different between SGCRL and Multi-goal CRL is how effectively they push representations for unsuccessful trajectory states away from the goal representation.
> >
> > However, this is not to say that lower dimensional representations are always better. For the rebuttal, we conducted additional ablations over representation dimension (Fig. 16 in **Appendix D.8**), which show that there is a minimum dimension required for orthogonality to hold. In practice, the optimal representation dimension is likely dependent on the complexity of the task.

---

### Official Review · Reviewer_8Y6w · 2025-10-31

**Soundness:** 3
**Presentation:** 2
**Contribution:** 2
**Rating:** 6
**Confidence:** 3

**Summary:**

The paper provides a mechanistic analysis of SGCRL [1], aiming to explain its “emergent exploration.” It formalizes the actor’s behavior as optimizing a shaped return based on similarity metrics, making explicit the intuition that policies are guided by goal-similarity in the learned representation rather than raw state-space distance. Building on this, it analyzes how InfoNCE training shapes the representation such that explored regions with low similarity to the goal are effectively “pruned”. The authors validate these predictions via controlled representation interventions and via a tabular reconstruction to show the exploratory behavior does not come from the use of function approximators. Experiments in grid worlds (e.g., Four Rooms) and toy tasks (e.g., Tower of Hanoi) empirically match the theory.

[1] Liu. et. al. A Single Goal is All You Need: Skills and Exploration Emerge from Contrastive RL without Rewards, Demonstrations, or Subgoals 2025

**Strengths:**

1. Clear mechanistic story linking the actor objective to an implicit ψ-similarity reward. Completing the intuition that SGCRL left as future work.

2. The use of interventions (attract/repel via embedding edits) and tabular models gives a good ablation study and nicely visualizes the behaviors.

3. Empirical trends in grid worlds align with the theory, making the presentation generally clean.

**Weaknesses:**

1. I'm concerned mostly about the scope mismatch: Framed as broadly “goal-conditioned RL,” but almost all analysis/evidence is for one algorithm (SGCRL) and mostly single-goal settings; multi-goal evidence is limited/toy and lacks HER-style goal-conditioned algorithms.

2. One of the main contributions, Theorem 1 helps with the formalization, but conceptually modest. It largely reframes the intuition as a theorem, but only under the alignment assumption.

3. Assumption of fixed ψ(g) limits the evidence strength. It does not hold using neural encoders (ψ(g) will drift). The paper shows robustness empirically, but the theoretical scope could be clearer.

4. Cognitive-science in the conclusion part is questionable. It reads as methodological borrowing (rational analysis, interventions, small model), not really a real "cognitive" experiment. I suggest this should be toned down or made more precise.

**Questions:**

1. Please explicitly scope when ψ(g) can be treated as fixed (tabular lookup, imaginary goal, frozen goal head) and discuss how in practice, the predictions degrade when ψ(g) drifts under shared encoders.

2. Can you provide multi-goal experiments with HER-style baselines (sampling goal from future states) to support the framing?

3. The sentence “drawn to high ψ-similarity even when far from the goal…” is confusing. Please state explicitly that attraction/repulsion is governed by similarity in embedding space, not raw state-space distance.

4. The role of the LogSumExp could potentially also be discussed in the represetation learning as it's so crucial to the contrastive RL. How does that impact the represetation/exploration?

---

> ### Author Response · Authors · 2025-11-20
>
> Dear Reviewer,
>
>
>
> Thank you for the thoughtful and detailed feedback. It seems like the main concerns relate to contextualizing SGCRL with multi-goal methods as well as the justification of theoretical assumptions. To address them, we have revised **Sections 2, 4.1, 4.2,** and **6 of the paper**, and we have added new experiments in **Appendices D.11** and **D.13**. All newly added content in the paper and appendix is highlighted in yellow. **Do the new analysis and experiments address the reviewer's concerns about the paper?**
>
>
> ---
>
>
>
> > multi-goal evidence is limited/toy and lacks HER-style goal-conditioned algorithms
>
> We have revised our **Related works** section to address this concern. To clarify the scope of our setting, we distinguish between **test-time** and **training-time** goals.
>
> In regards to test-time goals, our analysis and experiments focus on tasks which have a single hard goal at test time. These goals are often semantically meaningful (e.g. having all disks on the 3rd peg in Towers of Hanoi). Although our main analysis focuses on single-goal reaching tasks, we perform some experiments showing how the single-goal framework can be adapted to achieve *a distribution of test-time goals* while retaining the benefits of single-goal representation shaping (**Appendix D.7**).
>
>
> In regards to train-time goals, we want to clarify that SGCRL (and CRL more generally) is a HER-style goal-conditioned algorithm because it performs learning updates using future states relabeled as goals. The *single-goal* property refers to the fact that SGCRL is commanded only a single-goal (the test-time goal) during data collection. Previous work (Liu et al. 2025) showed that simply commanding this single-goal outperforms other HER-style baselines as well as a CRL baseline that employs a human-designed curriculum of data collection subgoals. Therefore, our contribution is to explain the surprising result of how *single-goal* data collection alone produces strong exploration. To make this distinction explicit, we added a third paragraph to the Related Work section clarifying our scope and positioning our contribution relative to prior multi-goal approaches. Moreover, we want to highlight **Appendix D.5**, which shows that *single-goal data collection shapes representations more effectively* than collecting data with a multi-goal distribution. This analysis provides insight into why SGCRL outperforms CRL with a manually-designed goal curriculum.
>
>
>
> ---
>
>
>
> > Theorem 1 ... reframes the intuition as a theorem, but only under the alignment assumption.
>
>
> We have revised **section 4.1** to address this concern. While we present the alignment requirement as **Assumption 1** in the main text for clarity, we want to highlight that the assumption is theoretically-motivated. **Appendix B.2.1** and **B.2.2** provide two derivations establishing concrete conditions under which the alignment property provably holds, thereby theoretically grounding Assumption 1 within practical settings. To keep the main exposition focused, we state the result as an assumption in the paper, while the appendix contains the full proofs and technical details.
>
>
> ---
>
>
>
> > Assumption of fixed ψ(g) limits the evidence strength ... Please explicitly scope when ψ(g) can be treated as fixed
>
>
> We have added a paragraph after **Theorem 2** that explains when the fixed-ψ(g) assumption is valid and what happens when ψ(g) drifts.
>
> We have added a new experiment in **Appendix D.11** testing Theorem 2 in the continuous-control setting with a **shared neural encoder**, where ψ(g) can drift as representations update. The experiment confirms that the orthogonality effect persists approximately, supporting the practical relevance of Theorem 2.
>
>
>
> ---
>
>
>
> > Cognitive-science in the conclusion ... reads as methodological borrowing
>
>
>
> Thanks for the suggestion for improving the presentation, which we have incorporated by revising the conclusion to clarify that our work **borrows methodological tools** from cognitive science rather than explicitly performing cognitive experiments. The revised conclusion now more accurately reflects the scope and contributions.
>
>
>
> ---
>
> > The sentence “drawn to high ψ-similarity even when far from the goal…” is confusing.
>
> We have revised this sentence.
>
>
> ---
>
> > The role of the LogSumExp
>
> We have added new experiments to address this concern. We would like to clarify that our main analysis uses the **InfoNCE backward loss** (Eq. 1), which does not include a LogSumExp term.
>
> For completeness, we add a new experiment in **Appendix D.13**, which analyzes analyzes the LogSumExp term in the **forward InfoNCE loss**. We find that the core representational dynamics remain consistent across both losses and using different logSumExp factors. We also tune the logSumExp coefficient and find that the default value yields fastest convergence, though all values yield similar performance.

---

### Author Response · Authors · 2025-12-01
**Discussion Summary**

Dear AC,

To help aid in the final decision, we provide a summary of our work and updates we made following the review discussion.

Our work studies Single-Goal Contrastive Reinforcement Learning (SGCRL), an unsupervised goal-conditional RL algorithm that exhibits emergent exploration. Our contribution is to improve interpretability of this algorithm through theoretical analysis and controlled experiments. Through these methods, we show that SGCRL implicitly shapes its representations to drive principled exploration.

All three initial reviews reccomended acceptance (score 6), citing "a clear mechanistic story" (8Y6w), "sound and novel analysis" of emergent exploration (GW6a), and "elegant and persuasive" use of tabular experiments (2Td2).

Reviewers 8Y6w and GW6a suggested that the work could be improved with additional analysis in for the multi-goal setting. To address this concern, we conducted two new experiments (Appendix D.14) where data collection during training uses different goals in each episode. In both cases, we observe that representations learned from **single-goal data collection** yield superior goal-directed exploration and avoid getting stuck in local minima.

To address concerns by Reviewer 2Td2 about connection to earlier ideas, we have revised Sections 4.1 and 4.2 to contextualize our contributions relative to successor representation literature. To address Reviewer 2Td2's concern about safety baselines, we have ran new safety baselines (PPO-LAG, CPO) on our safety task (Appendix D.12). We also add ablations over representation dimension, representation normalization and critic loss (Appendix D.8, D.9, D.10) to answer Reviewer 2Td2's questions about how these factors affect SGCRL dynamics.

To address concerns raised by Reviewer 8Y6w about assumptions for Theorem 2, we have added justification for Assumption 1 in Appendices B.2.1-B.2.2, clarified when the fixed-ψ(g) assumption holds, and conducted new experiments confirming that orthogonality holds with shared neural encoders that can drift (Appendix D.11).

We thank the reviewers for their thoughtful comments and engagement with the discussion. We have incorporated their suggestions to strengthen the paper. We refer the AC to the full discussion below for additional clarifications and discussion.

Kind Regards,

The Authors

---

### Meta-Review · Area_Chair_Ugru · 2026-01-04

**Summary:**

This work studies the mechanisms behind exploration in goal-conditioned reinforcement learning algorithms that learn contrastive representations. It talks about "emergent exploration", which is said to take place because such an agent implicitly maximizes an intrinsic reward defined by the similarity between each state and the goal state. As written by one of the reviewers: "This implicit reward gives rise to an automatic exploration–exploitation curriculum: states along unsuccessful trajectories gradually lose goal similarity (discouraging revisits), while those along successful paths gain similarity (encouraging exploitation)."

Ultimately, all reviewers saw value in the proposed contribution, and I personally do not think the paper should be evaluated on something outside of what it proposes to be (an investigation on single-goal RL). Thus, although I acknowledge this is a borderline paper given that it failed in getting any of the reviewers excited about it, I am recommending its acceptance.

**Reviewer Concerns:**

The reviewers were already recommending accepting the paper even before the authors had made any changes to the paper. The main concerns raised and comments about them are listed below:


- _Scope mismatch and framing: The paper is broadly scoped, but the actual results focus on a single algorithm. Similarly, other relevant work was not discussed._

	The authors have changed the presentation to better align the scope, and they have added some additional references and experiments in the Appendix. Reviewer GW6a still had concerns about the analysis added to Appendix D.14. However, I do think the authors have a good point when they point out the paper's main contribution is not an analysis between single-goal- vs multi-goal-conditioned reinforcement learning.

- _Theoretical results are somewhat limited due to their assumptions (e.g., alignment and fixed representation)_

	Additional results to motivate the alignment assumption were provided, and a discussion around the fixed/drift representation was also added.

- _The analysis inspired by cognitive science seems forceful._

	The authors have revised their conclusion.

**Reviewer Scores:**

I do not think any of the reviewers would have changed their final score. The paper would still end up with 6, 6, 6.

---

### Decision · Program_Chairs · 2026-01-26

Accept (Poster)